# Ellipsoidal Trust Region Methods for Neural Network Training

## Abstract

We investigate the use of ellipsoidal trust region constraints for second-order optimization of neural networks. This approach can be seen as a higher-order counterpart of adaptive gradient methods, which we here show to be interpretable as first-order trust region methods with ellipsoidal constraints. In particular, we show that the preconditioning matrix used in RMSProp and Adam satisfies the necessary conditions for provable convergence of second-order trust region methods with standard worst-case complexities. Furthermore, we run experiments across different neural architectures and datasets to find that the ellipsoidal constraints constantly outperform their spherical counterpart both in terms of number of backpropagations and asymptotic loss value. Finally, we find comparable performance to state-of-the-art first-order methods in terms of backpropagations, but further advances in hardware are needed to render Newton methods competitive in terms of time.

## 1 Introduction

We consider finite-sum optimization problems of the form

$$\min_{\mathbf{w} \in \mathbb{R}^d} \left[ \mathcal{L}(\mathbf{w}) := \sum_{i=1}^n \ell(f(\mathbf{w}, \mathbf{x}_i, \mathbf{y}_i)) \right], \tag{1}$$

which typically arise in neural network training, e.g. for empirical risk minimization over a set of data points $(\mathbf{x}_i, \mathbf{y}_i) \in \mathbb{R}^{in} \times \mathbb{R}^{out}, i = 1, \ldots, n$. Here, $\ell : \mathbb{R}^{out} \times \mathbb{R}^{out} \to \mathbb{R}^+$ is a convex loss function and $f : \mathbb{R}^{in} \times \mathbb{R}^d \to \mathbb{R}^{out}$ represents the neural network mapping parameterized by the concatenation of layers $\mathbf{w} \in \mathbb{R}^d$, which is non-convex due to its multiplicative nature and potentially non-linear activation functions. We assume that $\mathcal{L}$ is lower bounded and twice differentiable, i.e. $\mathcal{L} \in C^2(\mathbb{R}^d, \mathbb{R})$ and consider finding a first- and second-order stationary point $\bar{\mathbf{w}}$ for which $\|\nabla \mathcal{L}(\bar{\mathbf{w}})\| \leq \epsilon_g$ and $\lambda_{\min}\left(\nabla^2 \mathcal{L}(\bar{\mathbf{w}})\right) \geq -\epsilon_H$. Non-convex optimization problems are ubiquitous in machine learning. Among the most prominent examples are present-day deep neural networks, that have achieved outstanding results on core tasks such as collaborative filtering (Wang et al., 2015), sentence classification (Kim, 2014) and image classification (Krizhevsky et al., 2012).

In the era of big data and deep neural networks, stochastic gradient descent (SGD) is one of the most widely used training algorithms (Bottou, 2010). What makes SGD so attractive is its simplicity and per-iteration cost that are independent of the size of the training set ($n$) and scale linearly in the dimensionality ($d$). However, gradient descent is known to be inadequate to optimize functions that are ill-conditioned (Nesterov, 2013; Shalev-Shwartz et al., 2017) and thus adaptive gradient methods that employ dynamic, coordinate-wise learning rates based on past gradients—including Adagrad (Duchi et al., 2011), RMSprop (Tieleman & Hinton, 2012) and Adam (Kingma & Ba, 2014)—have become a popular alternative, often providing significant speed-ups over SGD. Yet, there exist no theoretical proofs that these methods are faster than gradient descent (Li & Orabona, 2018).

From a theoretical perspective, Newton methods provide stronger convergence guarantees by appropriately transforming the gradient in ill-conditioned regions according to second-order derivatives. It is precisely this Hessian information that allows *regularized* Newton methods to enjoy superlinear local convergence as well as to provably escape saddle points (Conn et al., 2000). While second-order algorithms have a long-standing history even in the realm of neural network training (Hagan & Menhaj, 1994; Becker et al., 1988), they were mostly considered as too computationally and memory

expensive for practical applications. Yet, the seminal work of Martens (2010) renewed interest for their use in deep learning by proposing efficient *Hessian-free* methods that only access second-order information via matrix-vector products which can be computed at the cost of an additional backpropagation (Pearlmutter, 1994; Schraudolph, 2002). Among the class of regularized Newton methods, trust region (Conn et al., 2000) and cubic regularization algorithms (Cartis et al., 2011) are the most principled approaches in the sense that they yield the strongest convergence guarantees. Recently, stochastic extensions have emerged (Xu et al., 2017b; Yao et al., 2018; Kohler & Lucchi, 2017; Gratton et al., 2017), which suggest their applicability for deep learning.

We here propose a simple modification to make TR methods even more suitable for neural network training. Particularly, we build upon the following alternative view on adaptive gradient methods:

*While gradient descent can be interpreted as a spherically constrained first-order TR method, preconditioned gradient methods—such as Adagrad—can be seen as first-order TR methods with ellipsoidal trust region constraint.*

This observation is particularly interesting since spherical constraints are blind to the underlying geometry of the problem, but ellipsoids can adapt to local landscape characteristics, thereby allowing for more suitable steps in regions that are ill-conditioned. We will leverage this analogy and investigate the use of the Adagrad and RMSProp preconditioning matrices as *ellipsoidal* trust region shapes within a stochastic second-order TR algorithm (Xu et al., 2017a; Yao et al., 2018). Since no ellipsoid fits all objective functions, our main contribution lies in the identification of adequate matrix-induced constraints that lead to provable convergence and significant practical speed-ups for the specific case of deep learning. On the whole, our contribution is threefold:

- We provide a new perspective on adaptive gradient methods that contributes to a better understanding of their inner-workings. Furthermore, we empirically find that many neural network problems exhibit diagonally dominated Hessian matrices which suggests the effectivity of *diagonal* preconditioning. (Section 3)
- We investigate the first application of ellipsoidal TR methods for deep learning. In Theorem 1 we show that the RMSProp matrix can directly be applied as constraint inducing norm in second-order TR algorithms while preserving all convergence guarantees (Theorem 2).
- Finally, we provide an experimental benchmark across different real-world datasets and architectures. We also compare against adaptive gradient methods and show results in terms of backprogations, epochs, and wall-clock time; a comparison we were not able to find in the literature. (Section 5)

Our main empirical results demonstrate that ellipsoidal constraints prove to be a very effective modification of the trust region method in the sense that they constantly outperform the spherical TR method, both in terms of number of backprogations and asymptotic loss value on a variety of tasks.

## 2 RELATED WORK

**First-order methods**    The prototypical method for optimizing Eq. (1) is SGD (Robbins & Monro, 1951). While the practical success of SGD in non-convex optimization is unquestioned, the theoretical foundation of this phenomenon is still rather limited. Recent findings suggest the ability of this method to escape saddle points and reach local minima in polynomial time for general non-convex problems, but they either need to artificially add noise to the iterates (Ge et al., 2015; Lee et al., 2016) or make an assumption on the inherent noise of vanilla SGD (Daneshmand et al., 2018). For neural network training, a recent line of research proclaims the effectiveness of SGD, but the results usually come at the cost of fairly strong assumptions such as heavy overparametrization and Gaussian inputs (Du et al., 2017; Brutzkus & Globerson, 2017; Li & Yuan, 2017; Du & Lee, 2018; Allen-Zhu et al., 2018). Adaptive gradient methods (Duchi et al., 2011; Tieleman & Hinton, 2012; Kingma & Ba, 2014) build on the intuition that larger learning rates for smaller gradient components and smaller learning rates for larger gradient components balance their respective influences and thereby make the methods behave as if they were optimizing a more isotropic surface. Such approaches have first been suggested for neural networks by LeCun et al. (2012). Recently, convergence guarantees for such methods are starting to appear (Ward et al., 2018; Li & Orabona, 2018). However, these are not superior to the $\mathcal{O}(\epsilon_g^{-2})$ worst-case complexity of standard gradient descent (Cartis et al., 2012b).

**Regularized Newton methods** The most principled class of regularized Newton methods are trust region (TR) and adaptive cubic regularization algorithms (ARC) (Conn et al., 2000; Cartis et al., 2011), which repeatedly optimize a local Taylor model of the objective while making sure that the step does not travel too far such that the model stays accurate. While the former finds first-order stationary points within $\mathcal{O}(\epsilon_g^{-2})$, ARC only takes at most $\mathcal{O}(\epsilon_g^{-3/2})$. However, simple modifications to the TR framework allow these methods to obtain the same accelerated rate (Curtis et al., 2017). Both methods take at most $\mathcal{O}(\epsilon_H^{-3})$ iterations to find an $\epsilon_H$ approximate second-order stationary point (Cartis et al., 2012a). These rates are optimal for second-order Lipschitz continuous functions (Carmon et al., 2017; Cartis et al., 2012a) and they can be retained even when only sub-sampled gradient and Hessian information is used (Kohler & Lucchi, 2017; Yao et al., 2018; Xu et al., 2017b; Blanchet et al., 2016; Liu et al., 2018; Cartis & Scheinberg, 2017). Furthermore, the involved Hessian information can be computed solely based on Hessian-vector products, which are implementable efficiently for neural networks (Pearlmutter, 1994). This makes these methods particularly attractive for deep learning, but the empirical evidence of their applicability is so far very limited. We are only aware of the works of Liu et al. (2018) and Xu et al. (2017a), which report promising first results but these are by no means fully encompassing.

**Gauss-Newton methods** An interesting line of research proposes to replace the Hessian by (approximations of) the generalized-Gauss-Newton matrix (GGN) within a Levenberg-Marquardt framework[1] (LeCun et al., 2012; Martens, 2010; Martens & Grosse, 2015). These methods have been termed *hessian-free* since only access to GGN-vector products is required. As the GGN matrix is always positive semidefinite, they cannot leverage negative curvature to escape saddles and hence, there exist no second-order convergence guarantees. Furthermore, there are cases in neural network training where the Hessian is better conditioned than the GGN matrix (Mizutani & Dreyfus, 2008). Nevertheless, the above works report promising preliminary results, most notably Grosse & Martens (2016) report that K-FAC can be faster than SGD on a small convnet. On the other hand, recent findings report performance at best comparable to SGD on the much larger ResNet architecture (Ma et al., 2019). Moreover, Xu et al. (2017a) reports many cases where TR and GGN algorithms perform similarly.

This line of work is to be seen as complementary to our approach since it is straight forward to replace the Hessian in the TR framework with the GGN matrix. Furthermore, the preconditioners used in Martens (2010) and Chapelle & Erhan (2011), namely diagonal estimates of the empirical Fisher and Fisher matrix, respectively, can directly be used as matrix norms in our ellipsoidal TR framework.

## 3 AN ALTERNATIVE VIEW ON ADAPTIVE GRADIENT METHODS

Adaptively preconditioned gradient methods update iterates as $\mathbf{w}_{t+1} = \mathbf{w}_t - \eta_t \mathbf{A}_t^{-1/2} \mathbf{g}_t$, where $\mathbf{g}_t$ is a stochastic estimate of $\nabla \mathcal{L}(\mathbf{w}_t)$ and $\mathbf{A}_t$ is a positive definite symmetric pre-conditioning matrix. In Adagrad, $\mathbf{A}_{ada,t}$ is the un-centered second moment matrix of the past gradients computed as

$$\mathbf{A}_{ada,t} := \mathbf{G}_t \mathbf{G}_t^\mathsf{T} + \epsilon \mathbf{I}, \tag{2}$$

where $\epsilon > 0$, $\mathbf{I}$ is the $d \times d$ identity matrix and $\mathbf{G}_t = [\mathbf{g}_1, \mathbf{g}_2, \ldots, \mathbf{g}_t]$. Building up on the intuition that past gradients might become obsolete in quickly changing non-convex landscapes, RMSprop (and Adam) introduce an exponential weight decay leading to the preconditioning matrix

$$\mathbf{A}_{rms,t} := \left( (1 - \beta) \mathbf{G}_t \operatorname{diag}(\beta^t, \ldots, \beta^0) \mathbf{G}_t^\mathsf{T} \right) + \epsilon \mathbf{I}, \tag{3}$$

where $\beta \in (0, 1)$. In order to save computational efforts, the diagonal versions $\operatorname{diag}(\mathbf{A}_{ada})$ and $\operatorname{diag}(\mathbf{A}_{rms})$ are more commonly applied in practice, which in turn gives rise to coordinate-wise adaptive stepsizes that are enlarged (reduced) in coordinates that have seen past gradient components with a smaller (larger) magnitude. In that way, the optimization methods can account for gradients of potentially different scales arising from e.g. different layers of the networks.

### 3.1 ADAPTIVE PRECONDITIONING AS ELLIPSOIDAL TRUST REGION

Starting from the fact that adaptive methods employ coordinate-wise stepsizes, one can take a principled view of these methods. Namely, their update steps arise from minimizing a first-order

---

[1]This algorithm is a simplified TR method, initially tailored for non-linear least squares problems (Nocedal & Wright, 2006)

Taylor model of the function $\mathcal{L}$ within an *ellipsoidal* search space around the current iterate $\mathbf{w}_t$, where the diameter of the ellipsoid along a particular coordinate is implicitly given by $\eta_t$ and $\|\mathbf{g}_t\|_{\mathbf{A}_t^{-1}}$. Correspondingly, vanilla (S)GD optimizes the same first-order model within a *spherical* constraint. Fig. 1 (top) illustrates this effect by showing not only the iterates of GD and Adagrad but also the implicit trust regions within which the local models were optimized at each step.[2] Since the models are linear, the constrained minimizer is always found on the boundary.

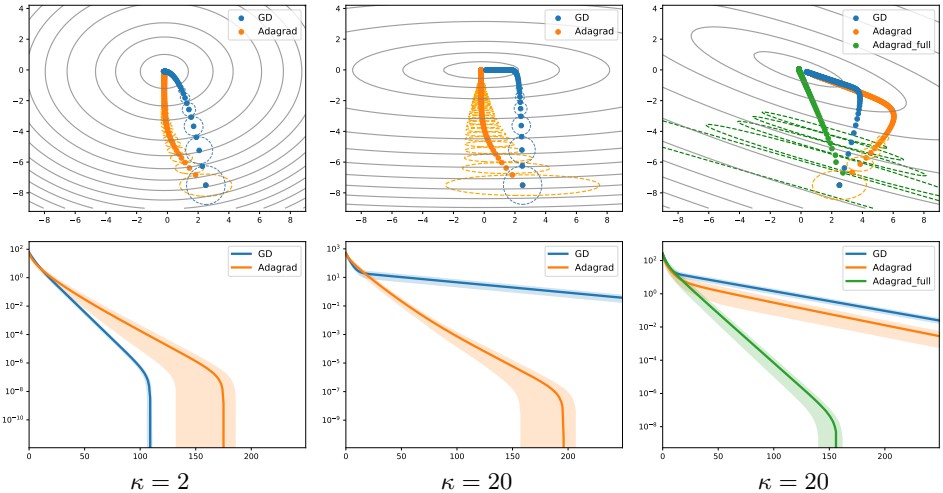

Figure 1: Top: Iterates and implicit trust regions of GD and Adagrad on three quadratic objectives with different condition number $\kappa$. Bottom: Average log suboptimality over iterations as well as 90% confidence intervals of 30 runs with random initialization

It is well known that GD struggles to progress towards the minimizer of quadratics along low-curvature directions (see e.g., Goh (2017)). While this effect is negligible for well-conditioned objectives (Fig. 1, left), it leads to drastically slow-down when the problem is ill-conditioned (Fig. 1, center). Particularly, once the method has reached the bottom of the valley, it struggles to make progress along the horizontal axis. Here is precisely where the advantage of adaptive stepsize methods comes into play. As illustrated by the dashed lines, Adagrad's search space is damped along the direction of high curvature (vertical axis) and elongated along the low curvature direction (horizontal axis). This allows the method to move further horizontally early on to enter the valley with a smaller distance to the optimizer $\mathbf{w}^*$ along the low curvature direction which accelerates convergence.

---

**Theorem 1** (Preconditioned gradient methods as TR). *A preconditioned gradient step*

$$\mathbf{w}_{t+1} - \mathbf{w}_t = \mathbf{s}_t := -\eta_t \mathbf{A}_t^{-1} \mathbf{g}_t \tag{4}$$

*with stepsize $\eta_t > 0$, symmetric positive definite preconditioner $\mathbf{A}_t \in \mathbb{R}^{d \times d}$ and $\mathbf{g}_t \neq 0$ minimizes a first-order model around $\mathbf{w}_t \in \mathbb{R}^d$ in an ellipsoid given by $\mathbf{A}_t$ in the sense that*

$$\mathbf{s}_t := \arg\min_{\mathbf{s} \in \mathbb{R}^d} \left[ m_t^1(\mathbf{s}) = \mathcal{L}(\mathbf{w}_t) + \mathbf{s}^\mathsf{T} \mathbf{g}_t \right], \qquad s.t. \quad \|\mathbf{s}\|_{\mathbf{A}_t} \leq \eta_t \|\mathbf{g}_t\|_{\mathbf{A}_t^{-1}}. \tag{5}$$

---

**Corollary 1** (Rmsprop). *The step $\mathbf{s}_{rms,t} := -\eta_t \mathbf{A}_{rms,t}^{-1/2} \mathbf{g}_t$ minimizes a first-order Taylor model around $\mathbf{w}_t$ in an ellipsoid given by $\mathbf{A}_{rms,t}^{1/2}$ (Eq. 3) in the sense that*

$$\mathbf{s}_{rms,t} := \arg\min_{\mathbf{s} \in \mathbb{R}^d} \left[ m_t^1(\mathbf{s}) = \mathcal{L}(\mathbf{w}_t) + \mathbf{s}^\mathsf{T} \mathbf{g}_t \right], \qquad s.t. \quad \|\mathbf{s}\|_{\mathbf{A}_{rms,t}^{1/2}} \leq \eta_t \|\mathbf{g}_t\|_{\mathbf{A}_{rms,t}^{-1/2}}. \tag{6}$$

Equivalent results can be established for Adam using $\mathbf{g}_{adam,t} := (1-\beta) \sum_{k=0}^t \beta^{t-k} \mathbf{g}_t$ as well as for Adagrad by replacing the matrix $\mathbf{A}_{ada}$ into the constraint in Eq. (6). Of course, the update procedure in Eq. (5) is merely a reinterpretation of the original preconditioned update, and thus the employed trust region radii are defined *implicitly* by the current gradient and stepsize.

---

[2]For illustrative purposes, we only plot every other trust region.

### 3.2 Diagonal versus full preconditioning

A closer look at Fig. 1 reveals that the first two problems come with level sets that are perfectly *axis-aligned*, which makes these objectives particularly attractive for diagonal preconditioning. For comparison, on the right of Fig. 1, we report another quadratic problem instance, where the Hessian is no longer zero on the off-diagonals. As can be seen, the interaction between coordinates introduces a tilt in the level sets and reduces the superiority of diagonal Adagrad over plain GD. However, using the full preconditioner $\mathbf{A}_{ada}$ re-establishes the original speed up. Yet, non-diagonal preconditioning comes at the cost of taking the inverse square root of a large matrix, which is why this approach has been relatively unexplored (see Agarwal et al. (2018) for a recent exception).

Interestingly, early results by Becker et al. (1988) on the curvature structure of neural nets report a strong diagonal dominance of the Hessian matrix $\nabla^2 \mathcal{L}(\mathbf{w})$. This suggests that the loss surface is indeed somewhat axis-aligned. However, the reported numbers are only for tiny feed-forward networks of at most 256 parameters. Therefore, we generalize these findings in the following to larger networks. Furthermore, we contrast the diagonal dominance of real Hessian matrices to the expected behavior of random Wigner matrices. Of course, true Hessians do not have i.i.d. entries but the symmetry of Wigner matrices suggests that this baseline is not completely off. Furthermore, we also compare Hessians of Ordinary Least Squares problems (OLS) with random inputs. For this purpose, let $\delta_{\mathbf{A}}$ define the ratio of diagonal to overall mass of a matrix $\mathbf{A}$, i.e. $\delta_{\mathbf{A}} := \frac{\sum_i |\mathbf{A}_{i,i}|}{\sum_i \sum_j |\mathbf{A}_{i,j}|}$ as in (Becker et al., 1988).

**Proposition 1** (Diagonal share of Wigner matrix). *For random Gaussian[3] Wigner matrix $\mathbf{W}$ formed as*

$$\mathbf{W}_{i,j} = \mathbf{W}_{j,i} := \begin{cases} \sim \mathcal{N}(0, \sigma_1^2), \ i < j \\ \sim \mathcal{N}(0, \sigma_2^2), \ i = j, \end{cases} \tag{7}$$

*where $\sim$ stands for i.i.d. draws (Wigner, 1993), the diagonal mass of the expected absolute matrix amounts to*

$$\delta_{\mathbb{E}[|\mathbf{W}|]} = \frac{1}{1 + (d-1)\frac{\sigma_2}{\sigma_1}}. \tag{8}$$

Thus, if we suppose the Hessian at any given point $\mathbf{w}$ were a random Wigner matrix we would expect the share of diagonal mass to fall with $\mathcal{O}(1/d)$ as the network grows in size. A similar result can be derived for the large $n$ limit in the case of OLS Hessians.

**Proposition 2** (Diagonal share of OLS Hessian). *Let $\mathbf{X} \in \mathbb{R}^{d \times n}$ and assume each $\mathbf{x}_{i,j}$ is generated i.i.d. with zero-mean finite second moment $\sigma^2 > 0$. Then the share of diagonal mass of the expected matrix $\mathbb{E}\left[|\mathbf{H}_{ols}|\right]$ amounts to*

$$\delta_{\mathbb{E}[|\mathbf{H}_{ols}|]} \overset{n \to \infty}{\to} \frac{\sqrt{n}}{\sqrt{n} + (d-1)\sqrt{\frac{2}{\pi}}}. \tag{9}$$

Again, we expect the diagonal mass to fall in $d$ and interestingly, empirical simulations suggest that the result of Proposition 2 holds already in small $n$ settings (see Fig.D.2) and it is likely that finite $n$ results can be derived when adding assumptions such as Gaussian data. As can be seen in Fig. 2 below, even for a practical batch size of $n = 32$ the diagonal mass $\delta_{\mathbf{H}}$ of neural networks stays above both benchmarks when its input is random as well as with real-world data at random initialization, during training and after convergence.

---

[3]The argument naturally extends to any distribution with positive expected absolute values.

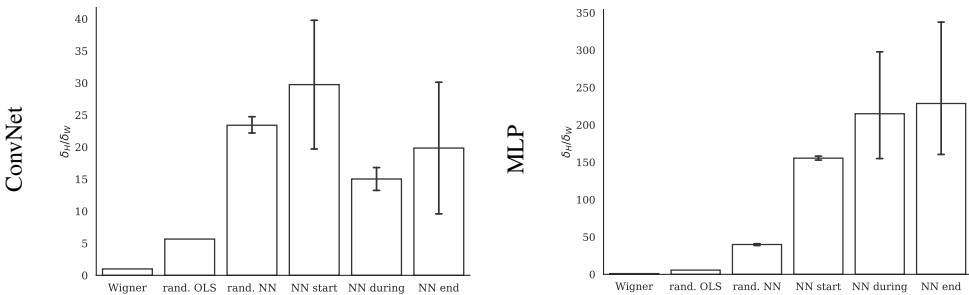

Figure 2: Diagonal mass of neural network Hessian $\delta_{\mathbf{H}}$ relative to $\delta_{\mathbb{E}[|\mathbf{W}|]}$ and $\delta_{\mathbb{E}[|\mathbf{H}_{\text{ols}}|]}$ of corresponding dimensionality for random inputs as well as at random initialization, middle and end of training with RMSProp on CIFAR-10. Mean and 95% CI over 10 independent runs.

These findings are in line with Becker et al. (1988) and suggest that full matrix preconditioning is most probably not worth the additional computational cost for neural networks. Consequently, we use diagonal preconditioning for both first- and second-order methods in all of our experiments in Section 5. Further theoretical elaboration of these findings present an interesting direction of future research.

## 4 SECOND-ORDER TRUST REGION METHODS

Cubic regularization (Nesterov & Polyak, 2006; Cartis et al., 2011) and trust region methods belong to the family of globalized Newton methods. Both frameworks compute parameter updates by optimizing regularized (former) or constrained (latter) second-order Taylor models of the objective $\mathcal{L}$ around the current iterate $\mathbf{w}_t$.[4] In particular, in iteration $t$ the update step of the trust region algorithm is computed as

$$\min_{\mathbf{s}\in\mathbb{R}^d} \left[ m_t(\mathbf{s}) := \mathcal{L}(\mathbf{w}_t) + \mathbf{g}_t^\mathsf{T}\mathbf{s} + \frac{1}{2}\mathbf{s}^\mathsf{T}\mathbf{B}_t\mathbf{s} \right], \qquad \text{s.t. } \|\mathbf{s}\|_{\mathbf{A}_t} \le \Delta_t \tag{10}$$

where $\Delta_t > 0$ and $\mathbf{g}_t$ and $\mathbf{B}_t$ are either $\nabla\mathcal{L}(\mathbf{w}_t)$ and $\nabla^2\mathcal{L}(\mathbf{w}_t)$ or suitable approximations. The matrix $\mathbf{A}_t$ induces the shape of the constraint set. So far, the common choice for neural networks is $\mathbf{A}_t := \mathbf{I}$, $\forall t$ which gives rise to spherical trust regions (Xu et al., 2017a; Liu et al., 2018). By solving the *constrained* problem (10), TR methods overcome the problem that pure Newton steps may be ascending, attracted by saddles or not even computable. See Appendix B for more details.

### 4.1 CONVERGENCE OF ELLIPSOIDAL TRUST REGION METHODS

Inspired by the success of adaptive gradient methods, we investigate the use of their preconditioning matrices as norm inducing matrices for second-order TR methods. The crucial condition for convergence is that the applied norms are not degenerate during the entire minimization process in the sense that the ellipsoids do not flatten out (or blow up) completely along any given direction. The following definition formalizes this intuition.

**Definition 1** (Uniformly equivalent norms). The norms $\|\mathbf{w}\|_{\mathbf{A}_t} := (\mathbf{w}^\mathsf{T}\mathbf{A}_t\mathbf{w})^{1/2}$ induced by symmetric positive definite matrices $\mathbf{A}_t$ are called uniformly equivalent, if $\exists\mu \ge 1$ such that

$$\frac{1}{\mu}\|\mathbf{w}\|_{\mathbf{A}_t} \le \|\mathbf{w}\|_2 \le \mu\|\mathbf{w}\|_{\mathbf{A}_t}, \qquad \forall\mathbf{w}\in\mathbb{R}^d, \forall t = 1, 2, \dots. \tag{11}$$

We now establish a result which shows that the RMSProp ellipsoid is indeed uniformly equivalent.

---

[4]In the following we only treat TR methods, but we would like to emphasize that the use of matrix induced norms can directly be transferred to the cubic regularization framework.

> **Lemma 1** (Uniform equivalence). *Suppose $\|\mathbf{g}_t\|^2 \leq L_H^2$ for all $\mathbf{w}_t \in \mathbb{R}^d$, $t = 1, 2, \ldots$ Then there always exists $\epsilon > 0$ such that the proposed preconditioning matrices $\mathbf{A}_{rms,t}$ (Eq. 3) are uniformly equivalent, i.e. Def. 1 holds. The same holds for the diagonal variant.*

Consequently, the ellipsoids $\mathbf{A}_{rms,t}$ can directly be applied to any convergent TR framework without losing convergence guarantees (Conn et al. (2000), Theorem 6.6.8).[5] Interestingly, this result cannot be established for $\mathbf{A}_{ada,t}$, which reflects the widely known vanishing stepsize problem that arises since squared gradients are continuously added to the preconditioning matrix. At least partially, this effect inspired the development of RMSprop (Tieleman & Hinton, 2012) and Adadelta (Zeiler, 2012).

**Why ellipsoids?**    There are many sources for ill-conditioning in neural networks such as un-centered and correlated inputs (LeCun et al., 2012), saturated hidden units, and different weight scales in different layers (Van Der Smagt & Hirzinger, 1998). While the quadratic term of model (10) accounts for such ill-conditioning to some extent, the spherical constraint is completely blind towards the loss surface. Thus, it is advisable to instead measure distances in norms that reflect the underlying geometry (see Chapter 7.7 in Conn et al. (2000)). The ellipsoids we propose are such that they allow for longer steps along coordinates that have seen small gradient components in past and vice versa. Thereby the TR shape is adaptively adjusted to fit the current region of the non-convex loss landscape. This procedure is not only effective when the iterates are in an ill-conditioned neighborhood of a minimizer (Figure 1), but it also helps to escape elongated plateaus (see autoencoder in Section 5).

### 4.2    A stochastic TR framework for neural network training

Since neural network training often constitutes a large-scale learning problem in which the number of datapoints $n$ is high, we here opt for a stochastic TR framework in order to circumvent memory issues and reduce the computational complexity. In particular, we adapt the framework of Yao et al. (2018); Xu et al. (2017b) to the case of iteration-dependent norm constraints (Algorithm 1). We prove that the $O\left(\max\left\{\epsilon_g^{-2}\epsilon_H^{-1}, \epsilon_H^{-3}\right\}\right)$ worst-case iteration complexity is retained, which is optimal if the function is second-order smooth (Cartis et al., 2012a). Towards this end, we assume that the derivative estimates are sufficiently accurate in the following sense.

**Assumption 1** (Sufficiently accurate derivatives). *The approximations of the gradient and Hessian at step $t$ satisfy*

$$\|\mathbf{g}_t - \nabla\mathcal{L}(\mathbf{w}_t)\| \leq \delta_g \text{ and } \|\mathbf{B}_t - \nabla^2\mathcal{L}(\mathbf{w}_t)\| \leq \delta_H,$$

*where $\delta_g \leq \frac{(1-\eta)\epsilon_g}{4}$ and $\delta_H \leq \min\left\{\frac{(1-\eta)v\epsilon_H}{2}, 1\right\}$, for some $0 < v < 1$.*

For finite-sum objectives such as Eq. 1, the above condition can be met by random sub-sampling due to classical concentration results for sums of random variables (Xu et al., 2017b; Kohler & Lucchi, 2017; Tripuraneni et al., 2017). Following these references, we assume access to the function value in each iteration for our theoretical analysis but we note that convergence can be retained even for fully stochastic trust region methods (Gratton et al., 2017; Chen et al., 2018; Blanchet et al., 2016) and indeed our experiments in Section 5 use sub-sampled function values due to memory constraints.

---

[5]Note that the assumption of bounded batch gradients, i.e. smooth objectives, is common in the analysis of stochastic algorithms (Allen-Zhu, 2017; Defazio et al., 2014; Schmidt et al., 2017; Duchi et al., 2011).

---

**Algorithm 1** Stochastic Ellipsoidal Trust Region Method

---

1: **Input:** $\mathbf{w}_0 \in \mathbb{R}^d, \gamma > 1, 1 > \eta > 0, \Delta_0 > 0$
2: **for** $t = 0, 1, \ldots,$ until convergence **do**
3:      Compute approximations $\mathbf{g}_t$ and $\mathbf{B}_t$. **If** $\|\mathbf{g}_t\| \leq \epsilon_g$, set $\mathbf{g}_t := 0$.
4:      Set $\mathbf{A}_t := \mathbf{A}_{rms,t}$ or $\mathbf{A}_t := \text{diag}\,(\mathbf{A}_{rms,t})$ (see Eq. (3)).
5:      Obtain $\mathbf{s}_t$ by solving $m_t(\mathbf{s}_t)$ approximately.
6:      Compute ratio of function over model decrease

$$\rho_t = \frac{\mathcal{L}(\mathbf{w}_t) - \mathcal{L}(\mathbf{w}_t + \mathbf{s}_t)}{m_t(\mathbf{0}) - m_t(\mathbf{s}_t)} \tag{12}$$

7:      Set

$$\Delta_{t+1} = \begin{cases} \gamma\Delta_t & \text{if } \rho_{\mathcal{S},t} > \eta \\ \Delta_t/\gamma & \text{if } \rho_{\mathcal{S},t} < \eta \end{cases}, \ \mathbf{w}_{t+1} = \begin{cases} \mathbf{w}_t + \mathbf{s}_t & \text{if } \rho_t \geq \eta \quad \text{(successful)} \\ \mathbf{w}_t & \text{otherwise} \quad \text{(unsuccessful)} \end{cases}$$

8: **end for**

---

**Theorem 2** (Convergence rate of Algorithm 1). *Assume that $\mathcal{L}(\mathbf{w})$ is second-order smooth with Lipschitz constants $L_g$ and $L_H$. Furthermore, let Assumption 1 and 2 hold. Then Algorithm 1 finds an $\mathcal{O}(\epsilon_g, \epsilon_H)$ first- and second-order stationary point in at most $\mathcal{O}\left(\max\left\{\epsilon_g^{-2}\epsilon_H^{-1}, \epsilon_H^{-3}\right\}\right)$ iterations.*

## 5 EXPERIMENTS

**Trust region methods** To validate our claim that ellipsoidal TR methods yield improved performance over spherical ones, we run a set of experiments on two image datasets and three types of network architectures. As can be seen in Figure 3, the ellipsoidal TR methods consistently outperform their spherical counterpart in the sense that they reach full training accuracy substantially faster on all problems. Moreover, their limit points are in all cases lower than those of the uniform TR method. Interestingly, this makes an actual difference in the image reconstruction quality of autoencoders (see Figure 12). We thus draw the clear conclusion that the ellipsoidal trust region constraints we propose are to be preferred over their spherical counterpart when training neural networks. Both the experimental and architectural details are provided in Appendix C.

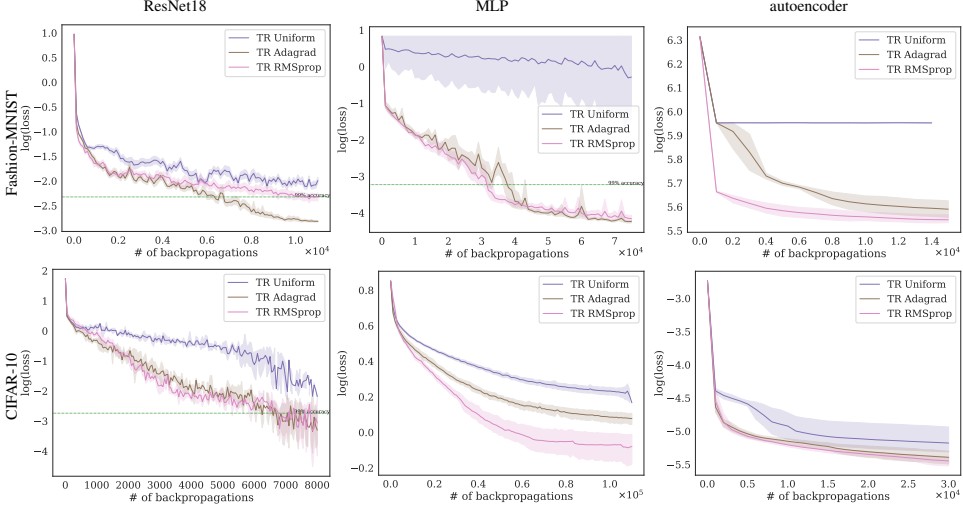

Figure 3: Log loss over backpropagations. Mean and 95% CI of 10 runs. Green dotted line indicates 99% acc.

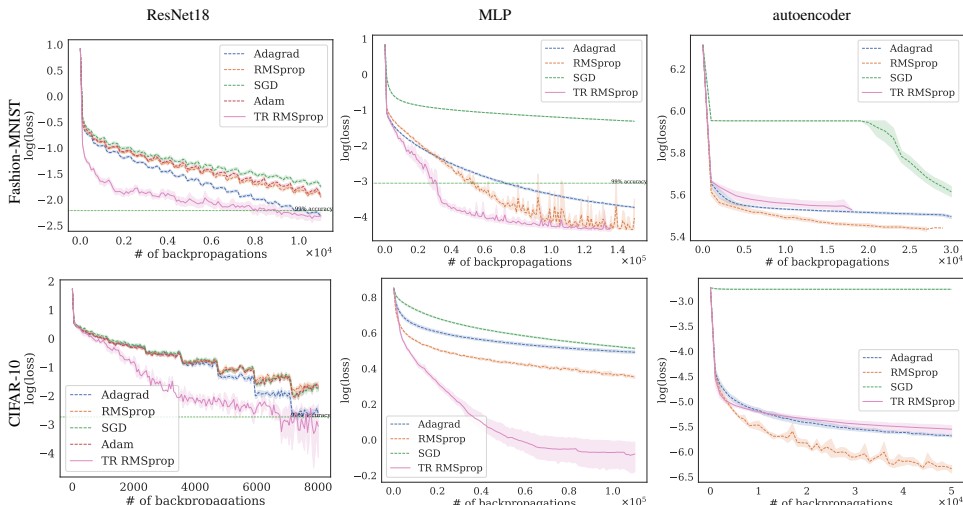

Figure 4: Log loss over backpropagations. Same setting as Figure 3. See Figure 10 for epoch results.

**Benchmark with SGD**   To put the previous results into context, we also benchmark several state-of-the-art gradient methods. We fix their sample size to 32 (as advocated e.g. in Masters & Luschi (2018)) but grid search the stepsize since it is the ratio of these two quantities that effectively determines the level of stochasticity (Jastrzebski et al., 2017). As the TR methods have a larger batch size[6] of 128–512, we report results both in terms of number of backpropagations and epochs for a fair comparison. A close look at Figure 4 and 10 (Appendix) indicates that the ellipsoidal TR methods can be slightly superior in terms of backprops but at best manage to keep pace with first-order methods in terms of epochs. Furthermore, the limit points of both first- and second-order methods yield the same order of loss in most experiments. When taking gradient norms into account (plot omitted), we indeed find no spurious local minima and only the autoencoders give rise to saddle points.

## 6   CONCLUSION

We investigated the use of ellipsoidal trust region constraints for neural networks. We have shown that the RMSProp matrix satisfies the necessary conditions for convergence and our experimental results demonstrate that ellipsoidal TR methods outperform their spherical counterparts significantly. We thus consider the development of further ellipsoids that can potentially adapt even better to the loss landscape such as e.g. (block-) diagonal hessian approximations (e.g. Bekas et al. (2007)) or approximations of higher order derivatives as an interesting direction of future research.

Yet, the gradient method benchmark indicates that the value of Hessian information for neural network training is limited for mainly three reasons: 1) second-order methods rarely yield better limit points, which suggests that saddles and spurious local minima are not a major obstacle; 2) gradient methods can run on smaller batch sizes which is beneficial in terms of epoch and when memory is limited; 3) The per-iteration time complexity is noticeably lower for first-order methods (Figure 11). These observations suggest that advances in hardware and distributed second-order algorithms (e.g., Osawa et al. (2018); Dünner et al. (2018)) will be needed before Newton-type methods can replace (stochastic) gradient methods in deep learning.

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

# Appendix A: Proofs

## A  NOTATION

Throughout this work, scalars are denoted by regular lower case letters, vectors by bold lower case letters and matrices as well as tensors by bold upper case letters. By $\|\cdot\|$ we denote an arbitrary norm. For a symmetric positive definite matrix $\mathbf{A}$ we introduce the compact notation $\|\mathbf{w}\|_{\mathbf{A}} = (\mathbf{w}^{\mathsf{T}}\mathbf{A}\mathbf{w})^{1/2}$, where $\mathbf{w} \in \mathbb{R}^d$.

## B  EQUIVALENCE OF PRECONDITIONED GRADIENT DESCENT AND FIRST-ORDER TRUST REGION METHODS

> **Theorem 3** (Theorem 1 restated). *A preconditioned gradient step*
>
> $$\mathbf{w}_{t+1} - \mathbf{w}_t = \mathbf{s}_t := -\eta_t \mathbf{A}_t^{-1}\mathbf{g}_t \tag{13}$$
>
> *with stepsize* $\eta_t > 0$*, symmetric positive definite preconditioner* $\mathbf{A}_t \in \mathbb{R}^{d \times d}$ *and* $\mathbf{g}_t \neq 0$ *minimizes a first-order local model around* $\mathbf{w}_t \in \mathbb{R}$ *in an ellipsoid given by* $\mathbf{A}_t$ *in the sense that*
>
> $$\mathbf{s}_t := \arg\min_{\mathbf{s} \in \mathbb{R}^d} \left[ m_t^1(\mathbf{s}) = \mathcal{L}(\mathbf{w}_t) + \mathbf{s}^{\mathsf{T}}\mathbf{g}_t \right], \quad s.t. \ \|\mathbf{s}\|_{\mathbf{A}_t} \leq \eta_t \|\mathbf{g}_t\|_{\mathbf{A}_t^{-1}}. \tag{14}$$

*Proof.* We start the proof by noting that the optimization problem in Eq. (14) is convex. For $\eta_t > 0$ the constraint satisfies the Slater condition since $0$ is a strictly feasible point. As a result, any KKT point is a feasible minimizer and vice versa.

Let $L(\mathbf{s}, \lambda)$ denote the Lagrange dual of Eq. (5)

$$L(\mathbf{s}, \lambda) := \mathcal{L}(\mathbf{w}_t) + \mathbf{s}^{\mathsf{T}}\mathbf{g}_t + \lambda \left( \|\mathbf{s}\|_{\mathbf{A}} - \eta_t\|\mathbf{g}_t\|_{\mathbf{A}_t^{-1}} \right). \tag{15}$$

Any point $\mathbf{s}$ is a KKT point if and only if the following system of equations is satisfied

$$\nabla_{\mathbf{s}} L(\mathbf{s}, \lambda) = \mathbf{g}_t + \frac{\lambda}{\|\mathbf{s}\|_{\mathbf{A}_t}} \mathbf{A}_t \mathbf{s} = 0 \tag{16}$$

$$\lambda \left( \|\mathbf{s}\|_{\mathbf{A}_t} - \eta_t\|\mathbf{g}_t\|_{\mathbf{A}_t^{-1}} \right) = 0. \tag{17}$$

$$\|\mathbf{s}\|_{\mathbf{A}_t} - \eta_t\|\mathbf{g}_t\|_{\mathbf{A}_t^{-1}} \leq 0 \tag{18}$$

$$\lambda \geq 0. \tag{19}$$

For $\mathbf{s}_t$ as given in Eq. (4) we have that

$$\|\mathbf{s}_t\|_{\mathbf{A}_t} = \sqrt{\eta_t^2 \mathbf{g}_t (\mathbf{A}_t^{-1})^{\mathsf{T}} \mathbf{A}_t \mathbf{A}_t^{-1} \mathbf{g}_t} = \eta_t \sqrt{\mathbf{g}_t \mathbf{A}_t^{-1} \mathbf{g}_t} = \eta_t\|\mathbf{g}_t\|_{\mathbf{A}_t^{-1}}. \tag{20}$$

and thus 17 and 18 hold with equality such that any $\lambda \geq 0$ is feasible. Furthermore,

$$\nabla_{\mathbf{s}} L(\mathbf{s}_t, \lambda) = \nabla f(\mathbf{w}_t) + \frac{\lambda}{\|\mathbf{s}_t\|_{\mathbf{A}_t}} \mathbf{A}_t \mathbf{s}_t \overset{(4)}{=} \mathbf{g}_t - \eta_t \frac{\lambda}{\eta_t\|\mathbf{g}_t\|_{\mathbf{A}_t^{-1}}} \mathbf{A}_t \mathbf{A}_t^{-1} \mathbf{g}_t = \mathbf{g}_t - \frac{\lambda}{\|\mathbf{g}_t\|_{\mathbf{A}_t^{-1}}} \mathbf{g}_t \tag{21}$$

is zero for $\lambda = \|\mathbf{g}_t\|_{\mathbf{A}^{-1}} \geq 0$. As a result, $\mathbf{s}_t$ is a KKT point of the convex problem 5 which proves the assertion.

$\square$

To illustrate this theoretical result we run gradient descent and Adagrad as well as the two corresponding first-order TR methods[7] on an ill-conditioned quadratic problem. While the method 1st TR

---

[7]Essentially Algorithm 1 with $m_t$ based on a *first* order Taylor expansion, i.e. $m_t^1(\mathbf{s})$ as in Eq. (14).

optimizes a linear model within a ball in each iteration, 1st TR$_{ada}$ optimizes the same model over the ellipsoid given by the Adagrad matrix $\mathbf{A}_{ada}$. The results in Figure 5 show that the methods behave very similar to their constant stepsize analogues.

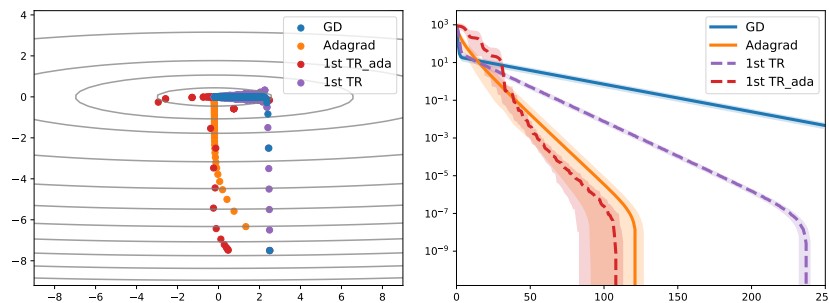

Figure 5: Iterates (left) and log suboptimality (right) of GD, Adagrad and two full-featured first-order TR algorithms of which one (1st TR) is spherically constraint and the other (1st TR$_{ada}$) uses $\mathbf{A}_{ada}$ as ellipsoid.

## C  CONVERGENCE OF ELLIPSOIDAL TR METHODS

In order to prove convergence results for ellipsoidal Trust Region methods one must ensure that the applied norms are coherent during the complete minimization process in the sense that the ellipsoids do not flatten out (or blow up) completely along any given direction. This intuition is formalized in Assumption 1 which we restate here for the sake of clarity.

**Definition 2** (Definition 1 restated). There exists a constant $\mu \geq 1$ such that

$$\frac{1}{\mu}\|\mathbf{w}\|_{\mathbf{A}_t} \leq \|\mathbf{w}\|_2 \leq \mu\|\mathbf{w}\|_{\mathbf{A}_t}, \qquad \forall t, \forall \mathbf{w} \in \mathbb{R}^d. \tag{22}$$

Towards this end, Conn et al. (2000) identify the following sufficient condition on the basis of which we will prove that our proposed ellipsoid $\mathbf{A}_{rms}$ is indeed uniformly equivalent under some mild assumptions.

**Lemma 2** (Theorem 6.7.1 in Conn et al. (2000)). *Suppose that there exists a constant $\zeta \geq 1$ such that*

$$\frac{1}{\zeta} \leq \sigma_{\min}(\mathbf{A}_t) \leq \sigma_{\max}(\mathbf{A}_t) \leq \zeta \qquad \forall t, \tag{23}$$

*then Definition 1 holds.*

Having uniformly equivalent norms is sufficient to prove convergence of ellipsoidal TR methods (se AN.1 and Theorem 6.6.8 in (Conn et al., 2000)). However, it is so far unknown how the ellipsoidal constraints influence the convergence *rate* itself. We here prove that the specific ellipsoidal TR method presented in Algorithm 1 preserves the rate of its spherically-constrained counterpart proposed in Yao et al. (2018) (see Theorem 2 below).

First, we show that the proposed $\mathbf{A}_{rms,t}$ ellipsoid satisfies Definition 1.

**Lemma 3** (Lemma 1). *Suppose $\|\mathbf{g}_t\|^2 \leq L_H^2$ for all $\mathbf{w}_t \in \mathbb{R}^d$, $t = 1, 2, \ldots$ Then there always exists $\epsilon > 0$ such that the proposed preconditioning matrices $\mathbf{A}_{rms,t}$ (Eq. 3) are uniformly equivalent, i.e. Def. 1 holds. The same holds for the diagonal variant.*

*Proof.* The basic building block of our ellipsoid matrix consists of the current and past stochastic gradients $\mathbf{G}_t := [\mathbf{g}_1, \mathbf{g}_2, \ldots, \mathbf{g}_t]$.

We consider $\mathbf{A}_{rms}$ which is built up as follows[8]

---

[8]This is a generalization of the diagonal variant proposed by Tieleman & Hinton (2012), which precondition the gradient step by an elementwise division with the square-root of the following estimate $g_t = (1 - \beta)g_{t-1} + \beta\nabla\mathcal{L}(\mathbf{w}_t)^2$.

$$\mathbf{A}_{rms,t} := \left( (1-\beta)\mathbf{G} \underbrace{\operatorname{diag}(\beta^t, \beta^{t-1}, \dots, \beta^0)}_{:=\mathbf{D}} \mathbf{G}^\mathsf{T} \right) + \epsilon\mathbf{I}. \tag{24}$$

From the construction of $\mathbf{A}_{rms,t}$ it directly follows that for any unit length vector $\mathbf{u} \in \mathbb{R}^d \setminus \{0\}, \|\mathbf{u}\|_2 = 1$ we have

$$
\begin{aligned}
&\mathbf{u}^\mathsf{T} \left( (1-\beta)\mathbf{G}\mathbf{D}\mathbf{G}^\mathsf{T} + \epsilon\mathbf{I} \right) \mathbf{u} \\
=&(1-\beta)\mathbf{u}^\mathsf{T}\mathbf{G}\mathbf{D}^{1/2}(\mathbf{D}^{1/2})^\mathsf{T}\mathbf{G}^\mathsf{T}\mathbf{u} + \epsilon\|\mathbf{u}\|_2^2 \\
=&(1-\beta)\left( (\mathbf{D}^{1/2})^\mathsf{T}\mathbf{G}^\mathsf{T}\mathbf{u} \right)^\mathsf{T} \left( (\mathbf{D}^{1/2})^\mathsf{T}\mathbf{G}^\mathsf{T}\mathbf{u} \right) + \epsilon\|\mathbf{u}\|_2^2 \\
\geq&\epsilon > 0,
\end{aligned}
\tag{25}
$$

which proves the lower bound for $\zeta = 1/\epsilon$. Now, let us consider the upper end of the spectrum of $\mathbf{A}_{rms,t}$. Towards this end, recall the geometric series expansion

$$\sum_{i=0}^{t} \beta^{t-i} = \sum_{i=0}^{t} \beta^i = \frac{1 - \beta^{t+1}}{1 - \beta} \tag{26}$$

and the fact that $\mathbf{G}\mathbf{G}^\top$ is a sum of exponentially weighted rank-one positive semi-definite matrices of the form $\mathbf{g}_i\mathbf{g}_i^\mathsf{T}$. Thus

$$\lambda_{max}(\mathbf{g}_i\mathbf{g}_i^\mathsf{T}) = \operatorname{Tr}(\mathbf{g}_i\mathbf{g}_i^\mathsf{T}) = \|\nabla \mathbf{g}_i\|^2 \leq L_H^2,$$

where the latter inequality holds per assumption for any sample size $|S|$. Combining these facts we get that

$$
\begin{aligned}
&\mathbf{u}^\mathsf{T} \left( (1-\beta)\mathbf{G}\mathbf{D}\mathbf{G}^\mathsf{T} + \epsilon\mathbf{I} \right) \mathbf{u} \\
=&(1-\beta)\mathbf{u}^\mathsf{T}\mathbf{G}\mathbf{D}\mathbf{G}^\mathsf{T}\mathbf{u} + \epsilon\|\mathbf{u}\|_2^2 \\
=&(1-\beta)\sum_{i=0}^{t} \beta^{t-1}\mathbf{u}^\mathsf{T}\mathbf{g}_i\mathbf{g}_i^\mathsf{T}\mathbf{u} + \epsilon\|\mathbf{u}\|_2^2 \\
\leq&(1-\beta)\sum_{i=0}^{t} \beta^{t-i}L_H^2\|\mathbf{u}\|_2^2 + \epsilon\|\mathbf{u}\|_2^2 \\
=&(1-\beta^{t+1})L_H^2 + \epsilon.
\end{aligned}
\tag{27}
$$

As a result we have that

$$\epsilon \leq \lambda_{min}(\mathbf{A}_{rms,t}) \leq \lambda_{max}(\mathbf{A}_{rms,t}) \leq \left(1 - \beta^{t+1}\right) L_H^2 + \epsilon \tag{28}$$

Finally, to achieve uniform equivalence we need the r.h.s. of (28) to be bounded by $1/\epsilon$. This gives rise to a quadratic equation in $\epsilon$, namely

$$\epsilon^2 + \left(1 - \beta^{t+1}\right) L_H^2\epsilon - 1 \leq 0 \tag{29}$$

which holds for any $t$ and any $\beta \in (0,1)$ as long as

$$0 \leq \epsilon \leq \frac{1}{2}(\sqrt{L_H^4 + 4} - L_H^2). \tag{30}$$

Such an $\epsilon$ always exists but one needs to choose smaller and smaller values as the upper bound on the gradient norm grows. For example, the usual value $\epsilon = 10^{-8}$ is valid for all $L_H^2 < 9.9 \cdot 10^7$. All of the above arguments naturally extend to the diagonal preconditioner $\operatorname{diag}(\mathbf{A}_{rms})$.

$\square$

Second, we note that it is no necessary to compute the update step by minimizing Eq. (10) to global optimality. Instead, it suffices to do better than the Cauchy- and Eigenpoint simultaneously (Conn et al., 2000; Yao et al., 2018). We here adapt this assumption for the case of iteration dependent norms (compare (Conn et al., 2000) Chapter 6). restate this assumption here

**Assumption 2** (Approximate model minimization). *Each update step* $\mathbf{s}_t$ *yields at least as much model decrease as the Cauchy- and Eigenpoint simultaneously, i.e.*

$$m_t(\mathbf{s}_t) \leq m_t(\mathbf{s}_t^C) \quad and \quad m_t(\mathbf{s}_t) \leq m_t(\mathbf{s}_t^E), \tag{31}$$

*where*

$$\mathbf{s}_t^C := \underset{0 \leq \alpha \leq \Delta_t}{\arg\min} \, m_t(-\alpha \frac{\mathbf{g}_t}{\|\mathbf{g}_t\|_t}) \quad and \quad \mathbf{s}_t^E := \underset{|\alpha| \leq \Delta_t}{\arg\min} \, m_t(\alpha \mathbf{u}_t), \tag{32}$$

*where* $\mathbf{u}_t$ *is an approximation to the corresponding negative curvature direction, i.e., for some* $0 < \nu < 1$, $\mathbf{u}_t^\intercal \mathcal{H}_t \mathbf{u}_t \leq \nu \left( \frac{\|\mathbf{u}_t\|}{\|\mathbf{u}_t\|_t} \right)^2 \lambda_{min}(\mathcal{B}_t)$ *and* $\|\mathbf{u}_t\|_t = 1$.

In practice, improving upon the Cauchy point is easily satisfied by any Krylov subspace method such as Conjugate Gradients, which ensures convergence to first order critical points. However, while the Steihaug-Toint CG solver can exploit negative curvature, it does not explicitly search for the most curved eigendirection and hence fails to guarantee $m_t(\mathbf{s}_t) \leq m_t(\mathbf{s}_t^E)$. Thus more elaborate Krylov descent methods such as Lanczos method might have to be employed for second-order criticality (See also Appendix B.2 and Conn et al. (2000) Chapter 7).

We now restate two results from Conn et al. (2000) that precisely quantify the model decrease guaranteed by Assumption 2.

**Lemma 4** (Model decrease: Cauchy Point (Theorem 6.3.1. in Conn et al. (2000))). *Suppose that* $\mathbf{s}_t^C$ *is computed as in Eq. (32). Then*

$$m_t(0) - m_t(\mathbf{s}_t^C) \geq \frac{1}{2} \|\mathbf{g}_t\| \min\{\frac{\|\mathbf{g}_t\|}{1 + \|\mathbf{B}_t\|}, \Delta_t \frac{\|\mathbf{g}_t\|}{\|\mathbf{g}_t\|_t}\}. \tag{33}$$

**Lemma 5** (Model decrease: Eigenpoint (Theorem 6.6.1 in Conn et al. (2000))). *Suppose that* $\lambda_{min}(\mathcal{B}_t) < 0$ *and* $\mathbf{s}_t^E$ *is computed as in Eq. (32). Then*

$$m_t(0) - m_t(\mathbf{s}_t^E) \geq -\frac{1}{2} \nu \lambda_{min}(\mathbf{B}_t) \left( \frac{\|\mathbf{u}_t\|}{\|\mathbf{u}_t\|_t} \right)^2 \Delta_t^2 \tag{34}$$

We are now ready to prove the final convergence results. Towards this end, we closely follow the line of arguments developed in Yao et al. (2018). First, we restate the following lemma which holds independent of the trust region constraint choice.

**Lemma 6** (Yao et al. (2018)). *Assume that* $\mathcal{L}(\mathbf{w})$ *is second-order smooth with Lipschitz constants* $L_g$ *and* $L_H$. *Furthermore, let Assumption 1 hold. Then*

$$F(\mathbf{x}_t + \mathbf{s}_t) - F(\mathbf{x}_t) - m_t(\mathbf{x}_t) \leq \mathbf{s}_t^\intercal \left( \nabla F(\mathbf{x}_t) - \mathbf{g}_t \right) + \frac{1}{2} \delta_H \|\mathbf{s}_t\|^2 + \frac{1}{2} L_H \|\mathbf{s}_t\|^3. \tag{35}$$

Second, we show that any iterate of Algorithm 1 is eventually successful as long as either the gradient norm or the smallest eigenvalue are above (below) the critical values $\epsilon_g$ and $\epsilon_H$ .

**Lemma 7** (Eventually successful iteration - $\|\mathbf{g}_t\| \geq \epsilon_g$). *Assume that* $\mathcal{L}(\mathbf{w})$ *is second-order smooth with Lipschitz constants* $L_g$ *and* $L_H$. *Furthermore, let Assumption 1 and 2 hold and suppose that* $\|\mathbf{g}_t\| \geq \epsilon_g$ *as well as*

$$\delta_g < \frac{1 - \eta}{4\mu^2} \epsilon_g, \ \Delta_t \leq \min \left\{ \frac{\mu \epsilon_g}{1 + L_g}, \sqrt{\frac{(1 - \eta)\epsilon_g}{12 L_H}} \frac{1}{\mu^4}, \frac{(1 - \eta)\epsilon_g}{3\mu^4} \right\}, \tag{36}$$

*then the step* $\mathbf{s}_t$ *is successful.*

*Proof.* First, by Assumption 2, Lemma 4, $\|\mathbf{g}_t\| \geq \epsilon_g$ and Lemma 1, we have

$$
\begin{aligned}
-m_t(\mathbf{s}_t) &\geq \frac{1}{2}\|\mathbf{g}_t\| \min\{\frac{\|\mathbf{g}_t\|}{1+\|\mathbf{B}_t\|}, \Delta_t \frac{\|\mathbf{g}_t\|}{\|\mathbf{g}_t\|_t}\} \\
&\geq \frac{1}{2}\|\mathbf{g}_t\| \min\{\frac{\epsilon_g}{1+\|\mathbf{B}_t\|}, \Delta_t \frac{\|\mathbf{g}_t\|}{\|\mathbf{g}_t\|_t}\} \\
&\geq \frac{1}{2}\|\mathbf{g}_t\| \min\{\frac{\epsilon_g}{1+\|\mathbf{B}_t\|}, \frac{\Delta_t}{\mu}\} \\
&= \frac{1}{2}\epsilon_g \frac{\Delta_t}{\mu},
\end{aligned}
\tag{37}
$$

where the last equality uses the above assumed upper bound on $\Delta_t$ of Eq. (36). Using this result together with Lemma 6 and the fact that $\|\mathbf{s}_t\|_2 \leq \mu\|\mathbf{s}_t\|_t \leq \mu\Delta_t$ due to Lemma 1, we find

$$
1 - \rho_t = \frac{\mathcal{L}(\mathbf{w}_t + \mathbf{s}_t) - \mathcal{L}(\mathbf{w}_t) - m_t(\mathbf{s}_t)}{-m_t(\mathbf{s}_t)} \tag{38}
$$

$$
\leq \frac{\delta_g \Delta_t \mu + \frac{1}{2}\delta_H \Delta_t^2 \mu^2 + \frac{1}{2}L_H \Delta_t^3 \mu^3}{\frac{1}{2}\epsilon_g \frac{\Delta_t}{\mu}} \tag{39}
$$

$$
= 2\frac{\delta_g}{\epsilon_g}\mu^2 + \frac{\delta_h}{\epsilon_g}\Delta_t \mu^3 + \frac{L_H}{\epsilon_g}\Delta_t^2 \mu^4 \tag{40}
$$

$$
\leq \frac{1-\eta}{2} + \left(\frac{\delta_H}{\epsilon_g}\Delta_t + \frac{L_H}{\epsilon_g}\Delta_t^2\right)\mu^4, \tag{41}
$$

where the last inequality makes use of the upper bound assumed on $\delta_g$. Now, we re-use the result of Lemma 10 in Yao et al. (2018), which states that $\left(\frac{\delta_H}{\epsilon_g}\Delta_t + \frac{L_H}{\epsilon_g}\Delta_t^2\right) \leq \frac{1-\eta}{2}$ for $\Delta_t \leq \min\left\{\sqrt{\frac{(1-\eta)\epsilon_g}{12L_H}}, \frac{(1-\eta)\epsilon_g}{3}\right\}$ to conclude that $\left(\frac{\delta_H}{\epsilon_g}\Delta_t + \frac{L_H}{\epsilon_g}\Delta_t^2\right)\mu^4 \leq \frac{1-\eta}{2}$ for our assumed bound on $\Delta_t$ in Eq. (36). As a result, Eq. (38) yields

$$
1 - \rho_t \leq 1 - \eta,
$$

which implies that the iteration $t$ is successful. $\qquad\square$

**Lemma 8** (Eventually successful iteration - $\lambda_{min}(\mathbf{B}_t) \leq -\epsilon_H$). *Assume that $\mathcal{L}(\mathbf{w})$ is second-order smooth with Lipschitz constants $L_g$ and $L_H$. Furthermore, let Assumption 1 and 2 hold and suppose that $\|\mathbf{g}_t\| < \epsilon_g$ and $\lambda_{min}(\mathbf{B}_t) < -\epsilon_H$. If*

$$
\delta_H < \frac{1-\eta}{2}\nu\epsilon_H, \Delta_t \leq \frac{(1-\eta)}{2\mu}\frac{\nu\epsilon_H}{L_H} \tag{42}
$$

*then iteration $t$ is successful.*

*Proof.* First, recall Eq. (35) and note that, since both $\mathbf{s}_t$ and $-\mathbf{s}_t$ are viable search directions, we can assume $\mathbf{s}_t^\mathsf{T}\nabla F(\mathbf{w}_t) \leq 0$ w.l.o.g.. Then

$$
\mathcal{L}(\mathbf{w}_t + \mathbf{s}_t) - \mathcal{L}(\mathbf{w}_t) - m_t(\mathbf{w}_t) \leq \frac{1}{2}\delta_H\|\mathbf{s}_t\|^2 + \frac{1}{2}L_H\|\mathbf{s}_t\|^3
$$

Therefore, recalling Eq. (34) as well as the fact that $\frac{\|\mathbf{u}_t\|_2}{\|\mathbf{u}_t\|_t} \leq \mu$ and $\|\mathbf{s}_t\|_2 \leq \mu\|\mathbf{s}_t\|_t \leq \mu\Delta_t$ due to Lemma 1

$$
\begin{aligned}
1 - \rho_t &= \frac{\mathcal{L}(\mathbf{w}_t + \mathbf{s}_t) - \mathcal{L}(\mathbf{w}_t) - m_t(\mathbf{s}_t)}{-m_t(\mathbf{s}_t)} \\
&\leq \frac{\frac{1}{2}\delta_H\|\mathbf{s}_t\|^2 + \frac{1}{2}L_H\|\mathbf{s}_t\|^3}{\frac{\nu}{2}|\lambda_{min}(\mathbf{B}_t)|\Delta_t^2\mu^2} \\
&\leq \frac{\frac{1}{2}\delta_H\|\mathbf{s}_t\|^2 + \frac{1}{2}L_H\|\mathbf{s}_t\|^3}{\frac{\nu}{2}\epsilon_H\Delta_t^2\mu^2} \\
&\leq \frac{\frac{1}{2}\delta_H\Delta_t^2\mu^2 + \frac{1}{2}L_H\Delta_t^3\mu^3}{\frac{\nu}{2}\epsilon_H\Delta_t^2\mu^2} \\
&= \frac{\delta_H}{\nu\epsilon_H} + \frac{L_H\Delta_t\mu}{\nu\epsilon_H} \\
&< 1 - \eta,
\end{aligned}
\tag{43}
$$

where the last second inequality is due to the conditions in Eq. (42). Therefore, $\rho_t \geq \eta$ and the iteration is successful. $\qquad\square$

Together, these two results allow us to establish a lower bound on the trust region radius $\Delta_t$.

**Lemma 9.** *Assume that $\mathcal{L}(\mathbf{w})$ is second-order smooth with Lipschitz constants $L_g$ and $L_H$. Furthermore, let Assumption 1 and 2 hold. Suppose*

$$
\delta_g < \frac{1-\eta}{4}\epsilon_g, \quad \delta_H < \min\{\frac{1-\eta}{2}\nu\epsilon_H, 1\}.
$$

*then for Algorithm 1 we have*

$$
\Delta_t \geq \frac{1}{\gamma}\min\left\{\frac{\epsilon_g\mu}{1+L_g}, \sqrt{\frac{(1-\eta)\epsilon_g}{12L_H\mu^8}}, \frac{(1-\eta)\epsilon_g}{3\mu^4}, \frac{(1-\eta)}{2\mu}\frac{\nu\epsilon_H}{L_H}\right\}, \quad \forall t = 1, 2, \ldots
\tag{44}
$$

*Proof.* The proof follows directly from $\Delta_t \geq \Delta_{t-1}/\gamma$ as well as the fact that any step is successful as soon as $\Delta_t$ falls below $\min\left\{\frac{\epsilon_g\mu}{1+L_g}, \sqrt{\frac{(1-\eta)\epsilon_g}{12L_H\mu^8}}, \frac{(1-\eta)\epsilon_g}{3\mu^4}, \frac{(1-\eta)}{2\mu}\frac{\nu\epsilon_H}{L_H}\right\}$ due to Lemma 7 and 8. $\quad\square$

**Lemma 10** (Number of successful iterations). *Under the same setting as Lemma 9, the number of successful iterations taken by Algorithm 1 is upper bounded by*

$$
|T_{succ}| \leq \frac{\mathcal{L}(\mathbf{x}_0) - \mathcal{L}(\mathbf{x}^*)}{C\epsilon_H\min\{\epsilon_g^2, \epsilon_H^2\}},
$$

*where* $C := \eta\min\{C_1, C_2\}$, $C_1 := \frac{1}{2}\min\left\{\frac{1}{1+L_g}, C_g\right\}$, $C_2 := \frac{\nu\mu^2}{2}\min\{C_g^2, C_H^2\}$, $C_g := \min\left\{\frac{\epsilon_g\mu}{1+L_g}, \sqrt{\frac{(1-\eta)\epsilon_g}{12L_H\mu^8}}, \frac{(1-\eta)\epsilon_g}{3\mu^4}\right\}$, $C_H := \frac{(1-\eta)}{2\mu}\frac{\nu\epsilon_H}{L_H}$

*Proof.* Suppose Algorithm 1 does not terminate at iteration $t$. Then either $\|\mathbf{g}_t\| \geq \epsilon_g$ or $\lambda_{min}(\mathbf{B}) \leq -\epsilon_h$. If $\|\mathbf{g}_t\| \geq \epsilon_g$, according to (33) and Lemma 1, we have

$$
\begin{aligned}
-m_t(\mathbf{s}_t) &\geq \frac{1}{2}\|\mathbf{g}_t\|\min\{\frac{\|\mathbf{g}_t\|}{1+\|\mathcal{H}_t\|}, \Delta_t\frac{1}{\mu}\} \\
&\geq \frac{1}{2}\epsilon_g\min\{\frac{\epsilon_g}{1+L_g}, C_g\epsilon_g, C_H\epsilon_H\} \\
&\geq C_1\epsilon_g\min\{\epsilon_g, \epsilon_H\}.
\end{aligned}
$$

Similarly, in the second case $\lambda_{min}(\mathbf{B}_t) \leq -\epsilon_h$, from Lemma 1 and 5 we have

$$
-m_t(\mathbf{s}_t) \geq \frac{1}{2}\nu|\lambda_{min}(\mathbf{B}_t)|\Delta_t^2\mu^2 \geq C_2\epsilon_H\min\{\epsilon_g^2, \epsilon_H^2\}.
$$

Let $T_{\text{succ}}$ denote the number of successful iterations. Since $\mathcal{L}(\mathbf{w})$ is monotonically decreasing, we have

$$
\begin{aligned}
\mathcal{L}(\mathbf{w}_0) - \mathcal{L}(\mathbf{w}^*) &\geq \sum_{t=0} \mathcal{L}(\mathbf{w}_t) - \mathcal{L}(\mathbf{w}_{t+1}) \\
&\geq \sum_{t \in T_{\text{succ}}} \mathcal{L}(\mathbf{w}_t) - \mathcal{L}(\mathbf{w}_{t+1}) \\
&\geq \sum_{t \in T_{\text{succ}}} -m_t(\mathbf{s}_t)\eta \\
&\geq \sum_{t \in T_{\text{succ}}} C\epsilon_H \min\{\epsilon_g^2, \epsilon_H^2\} \\
&\geq |T_{\text{succ}}| C\epsilon_H \min\{\epsilon_g^2, \epsilon_H^2\},
\end{aligned}
$$

which proves the assertion. □

We are now ready to prove the final result. Particularly, given the lower bound on $\Delta_t$ established in Lemma 9 we find an upper bound on the number of un-successful iterations, which combined with the result of Lemma 10 on the number of successful iterations yields the total iteration complexity of Algorithm 1.

---

**Theorem 4** (Theorem 2 restated). *Assume that $\mathcal{L}(\mathbf{w})$ is second-order smooth with Lipschitz constants $L_g$ and $L_H$. Furthermore, let Assumption 1 and 2 hold. Then Algorithm 1 finds an $\mathcal{O}(\epsilon_g, \epsilon_H)$ first- and second-order stationary point in at most $\mathcal{O}\left(\max\left\{\epsilon_g^{-2}\epsilon_H^{-1}, \epsilon_H^{-3}\right\}\right)$ iterations.*

---

*Proof.* The result follows by combining the lemmas 9 and 10 as in Theorem 1 of Xu et al. (2017a). □

## D    DIAGONAL DOMINANCE IN NEURAL NETWORKS

In the following, we make statements about the diagonal share of random matrices. As $\mathbb{E}[\frac{1}{x}]$ might not exist for a random variable $x$, we cannot compute the expectation of the diagonal share but rather of for computing the diagonal share of the expectation of the random matrix in absolute terms. Note that this notion is still meaningful, as the average of many non-diagonally dominated matrices with positive entries cannot become diagonally dominated.

### D.1    PROOF OF PROPOSITION 1

**Proposition 3** (Proposition 1 restated). *For random Gaussian Wigner matrix $\mathbf{W}$ formed as*

$$
\mathbf{W}_{i,j} = \mathbf{W}_{j,i} := \begin{cases} \sim \mathcal{N}(0, \sigma_1^2), \ i < j \\ \sim \mathcal{N}(0, \sigma_2^2), \ i = j, \end{cases} \tag{45}
$$

*where $\sim$ stands for i.i.d. draws (Wigner, 1993), the diagonal mass of the expected absolute matrix amounts to*

$$
\delta_{\mathbb{E}[|\mathbf{W}|]} = \frac{1}{1 + (d-1)\frac{\sigma_2}{\sigma_1}}. \tag{46}
$$

*Proof.*

$$
\begin{aligned}
\delta_{\mathbb{E}, \mathbf{W}} &= \frac{\sum_{k=1}^d \mathbb{E}\left[|\mathbf{W}_{k,k}|\right]}{\sum_{k=1}^d \sum_{l=1}^d \mathbb{E}\left[|\mathbf{W}_{k,l}|\right]} = \frac{d\mathbb{E}\left[|\mathbf{W}_{1,1}|\right]}{d\mathbb{E}\left[|\mathbf{W}_{1,1}|\right] + d(d-1)\mathbb{E}\left[|\mathbf{W}_{1,2}|\right]} \\
&= \frac{d\sigma_1\sqrt{2/\pi}}{d\sigma_1\sqrt{2/\pi} + d(d-1)\sigma_2\sqrt{2/\pi}} = \frac{1}{1 + \frac{d(d-1)\sigma_2\sqrt{2/\pi}}{d\sigma_1\sqrt{2/\pi}}} \\
&= \frac{1}{1 + (d-1)\frac{\sigma_2}{\sigma_1}}
\end{aligned} \tag{47}
$$

which simplifies to $\frac{1}{d}$ if the diagonal and off-diagonal elements come from the same Gaussian distribution ($\sigma_1 = \sigma_2$). □

For the sake of simplicity we only consider Gaussian Wigner matrices but the above argument naturally extends to any distribution with positive expected absolute values, i.e. we only exclude the Dirac delta function as probability density.

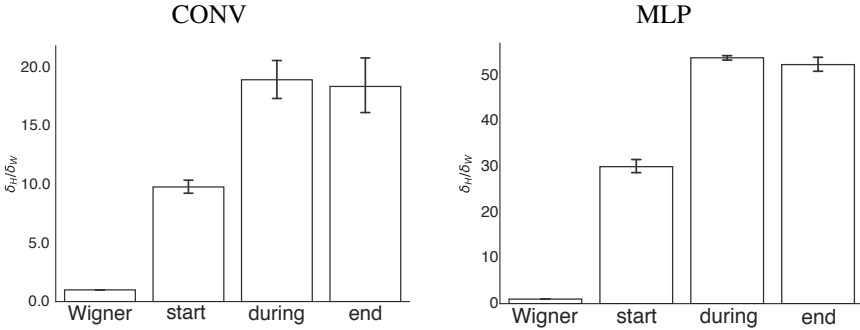

Figure 6: Share of diagonal mass of the Hessian $\delta_{\mathbf{H}}$ relative to $\delta_{\mathbf{W}}$ of the corresponding Wigner matrix at random initialization, after 50% iterations and at the end of training with RMSprop on MNIST. Average and 95% confidence interval over 10 runs. See Figure 2 for CIFAR-10 results.

## D.2 OLS BASELINE

When considering regression tasks, a direct competitor to neural network models is the classical Ordinary Least Squares (OLS) regression, which minimizes a quadratic loss over a *linear* model. In this case the Hessian simply amounts to the input-covariance matrix $\mathbf{H}_{ols} := \mathbf{X}^\intercal \mathbf{X}$, where $\mathbf{X} \in \mathbb{R}^{d \times n}$. We here show that the diagonal share of the expected matrix itself also decays in $d$, when $n$ grows to infinity. However, empirical simulations suggest the validity of this result even for much smaller values of $n$ (see Figure D.2) and it is likely that finite $n$ results can be derived when adding assumptions such as Gaussian data.

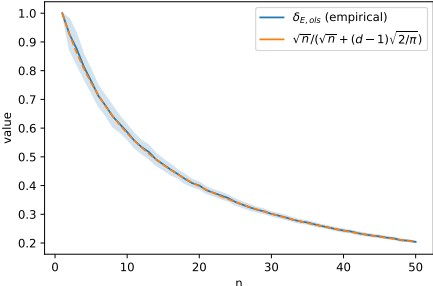

Figure 7: Validity of Proposition 2 in the small $n$ regime for Gaussian data.

**Proposition 4** (Proposition 2 restated). *Let* $\mathbf{X} \in \mathbb{R}^{n \times d}$ *and assume each* $\mathbf{x}_{i,k}$ *is generated i.i.d. with zero-mean finite second moment* $\sigma^2 > 0$. *Then the share of diagonal mass of the expected matrix* $\mathbb{E}\left[|\mathbf{H}_{ols}|\right]$ *amounts to*

$$\delta_{\mathbb{E}[|\mathbf{H}_{ols}|]} \overset{n \to \infty}{\to} \frac{\sqrt{n}}{\sqrt{n} + (d-1)\sqrt{\frac{2}{\pi}}} \tag{48}$$

*Proof.*

$$\delta_{\mathbb{E}[|\mathbf{H}_{\text{ols}}|]} = \frac{\sum_{k=1}^{d} \mathbb{E}\left[|(\mathbf{H}_{\text{ols}})_{k,k}|\right]}{\sum_{k=1}^{d}\sum_{l=1}^{d} \mathbb{E}\left[|(\mathbf{H}_{\text{ols}})_{k,l}|\right]} = \frac{\sum_{k=1}^{d} \mathbb{E}\left[|\sum_{i=1}^{n} \mathbf{x}_{i,k}^2|\right]}{\sum_{k=1}^{d}\sum_{l=1}^{d} \mathbb{E}\left[|\sum_{i=1}^{n} \mathbf{x}_{i,k}\mathbf{x}_{i,l}|\right]}$$

$$= \frac{d\sum_{i=1}^{n} \mathbb{E}\left[\mathbf{x}_{i,1}^2\right]}{d\sum_{i=1}^{n} \mathbb{E}\left[\mathbf{x}_{i,1}^2\right] + d(d-1)\mathbb{E}\left[|\sum_{i=1}^{n} \mathbf{x}_{i,1}\mathbf{x}_{i,2}|\right]} \tag{49}$$

Where we used the fact that all $\mathbf{x}_{i,k}$ are i.i.d. variables. Per assumption, we have $\mathbb{E}\left[\mathbf{x}_{i,1}^2\right] = \sigma^2, \forall i$. Furthermore, the products $\mathbf{x}_{i,1}\mathbf{x}_{i,2}$ are i.i.d with expectation 0 and variance $\sigma^4$. By the central limit theorem

$$Z_N = \frac{1}{\sqrt{n}}\sum_{i=1}^{n} \mathbf{x}_{i,1}\mathbf{x}_{i,2} \to Z$$

in law, with $Z \sim \mathcal{N}(0, \sigma^4)$, since $\mathbb{E}\left[Z_n^2\right] = \frac{1}{n}\mathbb{E}\left[\left(\sum_{i=1}^{n} \mathbf{x}_{i,1}\mathbf{x}_{i,2}\right)^2\right] = \frac{1}{n}\sum_{i=1}^{n} \mathbb{E}[\mathbf{x}_{i,1}^2]\mathbb{E}[\mathbf{x}_{i,2}^2] = \sigma^4$ due to the independence assumption. Then $E\left(|Z_N|\,1_{|Z_N| \geq R}\right) \leq E(|Z_N|^2/R) \leq \sigma^4/R$. This implies that

$$E\left(|Z_N|\right) \to E(|Z|) = \sqrt{\frac{2}{\pi}}\sigma^2$$

As a result, we have that in the limit of large $n$

$$\delta_{\mathbb{E}[|\mathbf{H}_{\text{ols}}|]} \overset{n\to\infty}{\to} \frac{dn\sigma^2}{dn\sigma^2 + d(d-1)\sqrt{n}\sqrt{\frac{2}{\pi}}\sigma^2} = \frac{1}{1 + \frac{(d-1)\sqrt{\frac{2}{\pi}}}{\sqrt{n}}} = \frac{\sqrt{n}}{\sqrt{n} + (d-1)\sqrt{\frac{2}{\pi}}} \tag{50}$$

$\square$

# Appendix B: Background on second-order optimization

## A  NEWTON'S METHOD

The canonical second-order method is Newton's methods. This algorithm uses the inverse Hessian as a scaling matrix and thus has updates of the form

$$\mathbf{w}_{t+1} = \mathbf{w}_t - \nabla^2 \mathcal{L}(\mathbf{w}_t)^{-1}\nabla\mathcal{L}(\mathbf{w}_t), \tag{51}$$

which is equivalent to optimizing the local quadratic model

$$m_N(\mathbf{w}_t) := \mathcal{L}(\mathbf{w}_t) + \nabla\mathcal{L}(\mathbf{w}_t)^\mathsf{T}\mathbf{s} + \frac{1}{2}\mathbf{s}^\mathsf{T}\nabla^2\mathcal{L}(\mathbf{w}_t)\mathbf{s} \tag{52}$$

to *first-order stationarity*. Using curvature information to rescale the steepest descent direction gives Newton's method the useful property of being linearly scale invariant. This gives rise to a *problem independent* local convergence rate that is super-linear and even quadratic in the case of Lipschitz continuous Hessians (see Nocedal & Wright (2006) Theorem 3.5), whereas gradient descent at best achieves linear local convergence (Nesterov, 2013).

However, there are certain drawbacks associated with applying classical Newton's method. First of all, the Hessian matrix may be singular and thus not invertible. Secondly, even if it is invertible the local quadratic model (Eq. 52) that is minimized in each NM iteration may simply be an inadequate approximation of the true objective. As a result, the Newton step is not necessarily a descent step. It may hence approximate arbitrary critical points (including local maxima) or even diverge. Finally, the cost of forming and inverting the Hessian sum up to $O(nd^2 + d^3)$ and are thus prohibitively high for applications in large dimensional problems.

# B    TRUST REGION METHODS

## B.1    OUTER ITERATIONS

Trust region methods are among the most principled approaches to overcome the above mentioned issues. These methods also construct a quadratic model $m_t$ but constrain the subproblem in such a way that the stepsize is restricted to stay within a certain radius $\Delta_t$ within which the model is trusted to be sufficiently adequate

$$\min_{s \in \mathbb{R}^d} m_t(\mathbf{s}) = \mathcal{L}(\mathbf{w}_t) + \nabla\mathcal{L}(\mathbf{w}_t)^\mathsf{T}\mathbf{s} + \frac{1}{2}\mathbf{s}^\mathsf{T}\nabla^2\mathcal{L}(\mathbf{w}_t)\mathbf{s}, \quad s.t. \; \|\mathbf{s}\| \leq \Delta_t. \tag{53}$$

Hence, contrary to line-search methods this approach finds the step $\mathbf{s}_t$ and its length $\|\mathbf{s}_t\|$ *simultaneously* by optimizing (53). Subsequently the actual decrease $\mathcal{L}(\mathbf{w}_t) - \mathcal{L}(\mathbf{w}_t + \mathbf{s}_t)$ is compared to the predicted decrease $m_t(0) - m_t(\mathbf{s}_t)$ and the step is only accepted if the ratio $\rho := \mathcal{L}(\mathbf{w}_t) - \mathcal{L}(\mathbf{w}_t + \mathbf{s}_t)/(m_t(0) - m_t(\mathbf{s}_t))$ exceeds some predefined success threshold $\eta_1 > 0$. Furthermore, the trust region radius is decreased whenever $\rho$ falls below $\eta_1$ and it is increased whenever $\rho$ exceeds the "very successful" threshold $\eta_2 0$. Thereby, the algorithm adaptively measures the accuracy of the second-order Taylor model – which may change drastically over the parameter space depending on the behaviour of the higher-order derivatives[9] – and adapts the effective length along which the model is trusted accordingly. See Conn et al. (2000) for more details.

As a consequence, the plain Newton step $\mathbf{s}_{N,t} = -\left(\nabla^2\mathcal{L}_t\right)^{-1}\nabla\mathcal{L}_t$ is only taken if it lies within the trust region radius and yields a certain amount of decrease in the objective value. Since many functions look somehow quadratic close to a minimizer the radius can be shown to grow asymptotically under mild assumptions such that eventually full Newton steps are taken in every iteration which retains the local quadratic convergence rate (Conn et al., 2000).

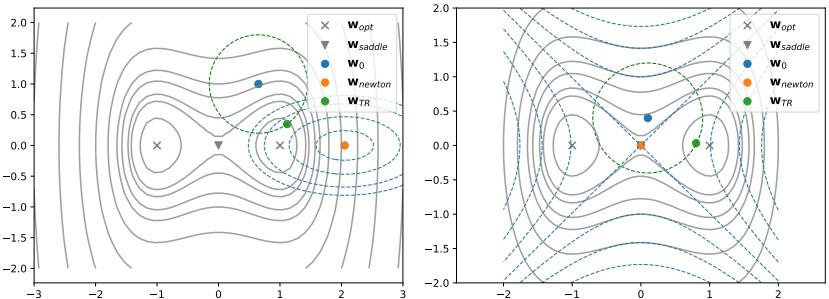

Figure 8: Level sets of the non-convex, coercive objective function $f(\mathbf{w}) = 0.5\mathbf{w}_0^2 + 0.25\mathbf{w}_1^4 - 0.5\mathbf{w}_1^2$. Newton's Method makes a local quadratic model (blue dashed lines) and steps to its *critical point*. It may be thus be ascending (left) or attracted by a saddle point (right). TR methods relieve this issue by stepping to the *minimizer* of that model *within* a certain region (green dashed line).

## B.2    SUBPROBLEM SOLVER

Interestingly, there is no need to optimize Eq. (53) to global optimality to retain the remarkable global convergence properties of TR algorithms. Instead, it suffices to do better than the Cauchy- and Eigenpoint[10] simultaneously. One popular approach is to minimize $m_t(\mathbf{s})$ in nested Krylov subspaces. These subspaces naturally include the gradient direction as well as increasingly accurate estimates of the leading eigendirection

$$\text{span}\{\mathbf{g}_t, \mathbf{B}_t\mathbf{g}_t, \mathbf{B}_t^2\mathbf{g}_t, \ldots, \mathbf{B}_t^j\mathbf{g}_t\} \tag{54}$$

until (for example) the stopping criterion

$$\|\nabla m_t(\mathbf{s}_j)\| \leq \|\nabla\mathcal{L}(\mathbf{w}_t)\| \min\{\kappa_K, \|\nabla\mathcal{L}(\mathbf{w}_t)\|^\theta\}, \quad \kappa_K < 1, \theta \geq 0 \tag{55}$$

---

[9]Note that the second-order Taylor models assume constant curvature.

[10]which are the model minimizers along the gradient and the eigendirection associated with its smallest eigenvalue, respectively.

is met, which requires increased accuracy as the underlying trust region algorithm approaches criticality. Conjugate gradients and Lanczos method are two iterative routines that implicitly build up a conjugate and orthogonal basis for such a Krylov space respectively and they converge *linearly* on quadratic objectives with a square-root dependency on the condition number of the Hessian (Conn et al., 2000). We here employ the preconditionied Steihaug-Toint CG method (Steihaug, 1983) in order to cope with possible boundary solutions of (53) but similar techniques exist for the Lanczos solver as well for which we also provide code. As preconditioning matrix for CG we use the same matrix as for the ellipsoidal constraint.

## C  DAMPED (GAUSS-)NEWTON METHODS

An alternative approach to actively constraining the region within which the model is trusted is to instead penalize the step norm in each iteration in a Lagrangian manner. This is done by so-called damped Newton methods that add a $\lambda > 0$ multiple of the identity matrix to the second-order term in the model, which leads to the update step

$$
\begin{aligned}
\min_{s \in \mathbb{R}^d} m_t(\mathbf{s}) &= \mathcal{L}(\mathbf{w}_t) + \nabla \mathcal{L}(\mathbf{w}_t)^{\mathsf{T}} \mathbf{s} + \frac{1}{2} \mathbf{s}^{\mathsf{T}} (\nabla^2 \mathcal{L}(\mathbf{w}_t) + \lambda \mathbf{I}) \mathbf{s} \\
&= \mathcal{L}(\mathbf{w}_t) + \nabla \mathcal{L}(\mathbf{w}_t)^{\mathsf{T}} \mathbf{s} + \frac{1}{2} \mathbf{s}^{\mathsf{T}} \nabla^2 \mathcal{L}(\mathbf{w}_t) \mathbf{s} + \lambda \|\mathbf{s}\|^2.
\end{aligned}
\tag{56}
$$

This can also be solved hessian-free by conjugate gradients (or other Krylov subspace methods). The penalty parameter $\lambda$ is acting inversely to the trust region radius $\Delta$ and it is often updated accordingly. Such algorithms are commonly known as Levenberg-Marquardt algorithms and they were originally tailored towards solving non-linear least squares problems (Nocedal & Wright, 2006) but they have been proposed for neural network training already early on (Hagan & Menhaj, 1994).

Many algorithms in the existing literature replace the use of $\nabla^2 \mathcal{L}(\mathbf{w}_t)$ in (56) with the Generalized Gauss Newton matrix (Martens, 2010; Chapelle & Erhan, 2011) or an approximation of the latter (Martens & Grosse, 2015). This matrix constitutes the first part of the well-known Gauss-Newton decomposition

$$
\nabla^2 \mathcal{L}(\cdot) = \underbrace{\frac{1}{n} \sum_{i=1}^{n} \ell''(f_i(\cdot)) \nabla f_i(\cdot) \nabla f_i(\cdot)^{\mathsf{T}}}_{:= \mathbf{A}_{GGN}} + \frac{1}{n} \sum_{i=1}^{n} \ell'(f_i(\cdot)) \nabla^2 f_i(\cdot),
\tag{57}
$$

where $l'$ and $l''$ are the first and second derivative of $l : \mathbb{R}^{out} \to \mathbb{R}^+$ assuming that $out = 1$ (binary classification and regression task) for simplicity here.

It is interesting to note that the GGN matrix $\mathbf{A}_{GGN}$ of neural networks is equivalent to the Fisher matrix used in natural gradient descent (Amari, 1998) in many cases like linear activation function and squared error as well as sigmoid and cross-entropy or softmax and negative log-likelihood for which the extended Gauss-Newton is defined (Pascanu & Bengio, 2013). As can be seen in (57) the matrix $\mathbf{A}_{GGN}$ is positive semidefinite (and low rank if $n < d$). As a result, there exist no second-order convergence guarantees for such methods on general non-convex problems. On the other end of the spectrum, the GGN also drops possibly positive terms from the Hessian (see 57). Hence it is not guaranteed to be an upper bound on the latter in the PSD sense. Essentially, GGN approximations assume that the network is piece-wise linear and thus the GGN and Hessian matrices only coincide in the case of linear and ReLU activations or non-curved loss functions. For any other activation the GGN matrix may approximate the Hessian only asymptotically and if the $\ell'(f_i(\cdot))$ terms in 57 go to zero for all $i \in \{1, \ldots, n\}$. In non-linear least squares such problems are called zero-residual problems and GN methods can be shown to have quadratic local convergence there. In any other case the convergence rate does not exceed the linear local convergence bound of gradient descent. In practice however there are cases where deep neural nets do show negative curvature in the neighborhood of a minimizer (Bottou et al., 2018).Finally, Dauphin et al. (2014) propose the use of the absolute Hessian instead of the GGN matrix in a framework similar to 56. This method has been termed *saddle-free Newton* even though its manifold of attraction to a given saddle is non-empty[11].

---

[11]It is the same as that for GD, which renders the method unable to escape e.g. when initialized right on a saddle point. To be fair, the manifold of attraction for GD constitutes a measure zero set (Lee et al., 2016).

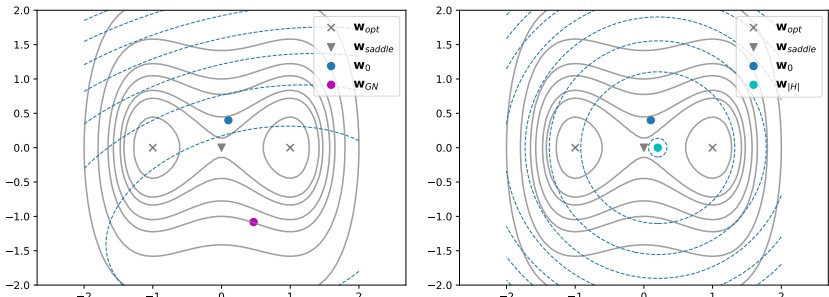

Figure 9: Both, the GGN method and saddle-free Newton method make a positive definite quadratic model around the current iterate and thereby overcome the abstractedness of pure Newton towards the saddle (compare Figure 8). However, (i) none of these methods can escape the saddle once they are in the gradient manifold of attraction and (ii) as reported in Mizutani & Dreyfus (2008) the GN matrix can be significantly less well conditioned than the absolute Hessian (here $\kappa_{GN} = 49'487'554$ and $\kappa_{|H|} = 1.03$ so we had to add a damping factor of $\lambda = 0.1$ to make the GN step fit the plot.

## C.1 COMPARISON TO TRUST REGION

*Contrary* to TR methods, the Levenberg-Marquardt methods never take plain Newton steps since the regularization is always on ($\lambda > 0$). Furthermore, if a positive-definite Hessian approximation like the Generalized Gauss Newton matrix is used, this algorithm is not capable of exploiting negative curvature and there are cases in neural network training where the Hessian is much better conditioned than the Gauss-Newton matrix (Mizutani & Dreyfus, 2008) (also see Figure 9). While some scholars believe that positive-definiteness is a desirable feature (Martens, 2010; Chapelle & Erhan, 2011), we want to point out that following negative curvature directions is necessarily needed to escape saddle points and it can also be meaningful to follow directions of negative eigenvalue $\lambda$ outside a saddle since they guarantee $\mathcal{O}(|\lambda|^3)$ progress, whereas a gradient descent step yields at least $\|\nabla f(\mathbf{w})\|^2$ progress (both under certain stepsize conditions) and one cannot conclude a-priori which one is better in general (Curtis & Robinson, 2017; Alain et al., 2018). Despite these theoretical considerations, many methods based on GGN matrices have been applied to neural network training (see Martens (2014) and references therein) and particularly the hessian-free implementations of (Martens, 2010; Chapelle & Erhan, 2011) can be implemented very cheaply (Schraudolph, 2002).

## D USING HESSIAN INFORMATION IN NEURAL NETWORKS

While many theoretical arguments suggest the superiority of regularized Newton methods over gradient based algorithms, several practical considerations cast doubt on this theoretical superiority when it comes to neural network training. Answers to the following questions are particularly unclear: Are saddles even an issue in deep learning? Is superlinear local convergence a desirable feature in machine learning applications (test error)? Are second-order methods more "vulnerable" to sub-sampling noise? Do worst-case iteration complexities even matter in real-world settings? As a result, the value of Hessian information in neural network training is somewhat unclear a-priori and so far a conclusive empirical study is still missing.

Our empirical findings indicate that the net value of Hessian information for neural network training is indeed somewhat limited for mainly three reasons: 1) second-order methods rarely yield better limit points, which suggests that saddles and spurious local minima are not a major obstacle; 2) gradient methods can indeed run on smaller batch sizes which is beneficial in terms of epoch and when memory is limited; 3) The per-iteration time complexity is noticeably lower for first-order methods. In summary, these observations suggest that advances in hardware and distributed second-order algorithms (e.g., Osawa et al. (2018); Dünner et al. (2018)) will be needed before Newton-type methods can replace (stochastic) gradient methods in deep learning.

# Appendix C: Experiment details

## A EXPERIMENTAL RESULTS

### A.1 ELLIPSOIDAL TRUST REGION VS. FIRST-ORDER OPTIMIZER

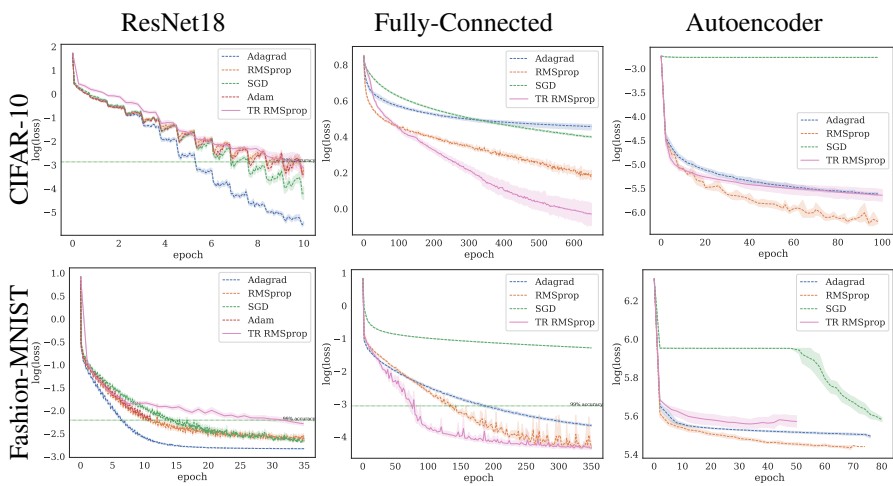

Figure 10: Experiment comparing TR and gradient methods in terms of epochs. Average log loss as well as 95% confidence interval shown.

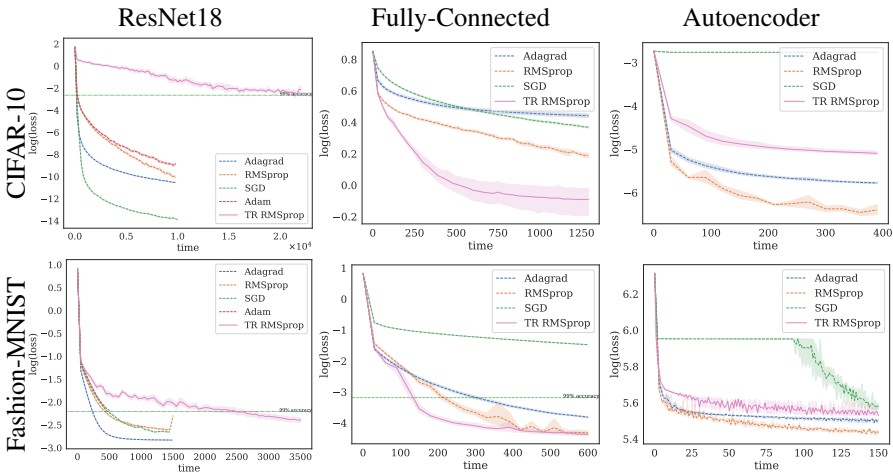

Figure 11: Experiment comparing TR and gradient methods in terms of wall-clock time. Average log loss as well as 95% confidence interval shown. The advantage of extremely low-iteration costs of first-order methods is particularly notable in the ResNet18 architecture due to the large network size.

## B FURTHER EXPERIMENT DETAILS

### B.1 DEFAULT PARAMETERS, ARCHITECTURES AND DATASETS

**Parameters** Table 1 reports the default parameters we consider. Only for the larger ResNet18 on CIFAR-10, we adapted the batch size to 128 due to memory constraints.

| | $|\mathcal{S}_0|$ | $\Delta_0$ | $\Delta_{\max}$ | $\eta_1$ | $\eta_2$ | $\gamma_1$ | $\gamma_2$ | $\kappa_K$ (krylov tol.) |
|---|---|---|---|---|---|---|---|---|
| $TR_{uni}$ | 512 | $10^{-4}$ | 10 | $10^{-4}$ | 0.95 | 1.1 | 1.5 | 0.1 |
| $TR_{ada}$ | 512 | $10^{-4}$ | 10 | $10^{-4}$ | 0.95 | 1.1 | 1.5 | 0.1 |
| $TR_{rms}$ | 512 | $10^{-4}$ | 10 | $10^{-4}$ | 0.95 | 1.1 | 1.75 | 0.1 |

Table 1: Default parameters

**Datasets** We use two real-world datasets for image classification, namely CIFAR-10 and Fashion-MNIST[12]. While Fashion-MNIST consists of greyscale $28 \times 28$ images, CIFAR-10 are colored images of size $32 \times 32$. Both datasets have a fixed training-test split consisting of 60,000 and 10,000 images, respectively.

**Network architectures** The MLP architectures are simple. For MNIST and Fashion-MNIST we use a $784 - 128 - 10$ network with tanh activations and a cross entropy loss. The networks has $101'770$ parameters. For the CIFAR-10 MLP we use a $3072 - 128 - 128 - 10$ architecture also with tanh activations and cross entropy loss. This network has $410'880$ parameters.

The Fashion-MNIST autoencoder has the same architecture as the one used in Hinton & Salakhutdinov (2006); Xu et al. (2017a); Martens (2010); Martens & Grosse (2015). The encoder structure is $784 - 1000 - 500 - 250 - 30$ and the decoder is mirrored. Sigmoid activations are used in all but the central layer. The reconstructed images are fed pixelwise into a binary cross entropy loss. The network has a total of $2'833'000$ parameters. The CIFAR-10 autoencoder is taken from the implementation of https://github.com/jellycsc/PyTorch-CIFAR-10-autoencoder.

For the ResNet18, we used the implementation from torchvision for CIFAR-10 as well as a modification of it for Fashion-MNIST that adapts the first convolution to account for the single input channel.

In all of our experiments each method was run on one Tesla P100 GPU using the PyTorch (Paszke et al., 2017) library.

## B.2 RECONSTRUCTED IMAGES FROM AUTOENCODERS

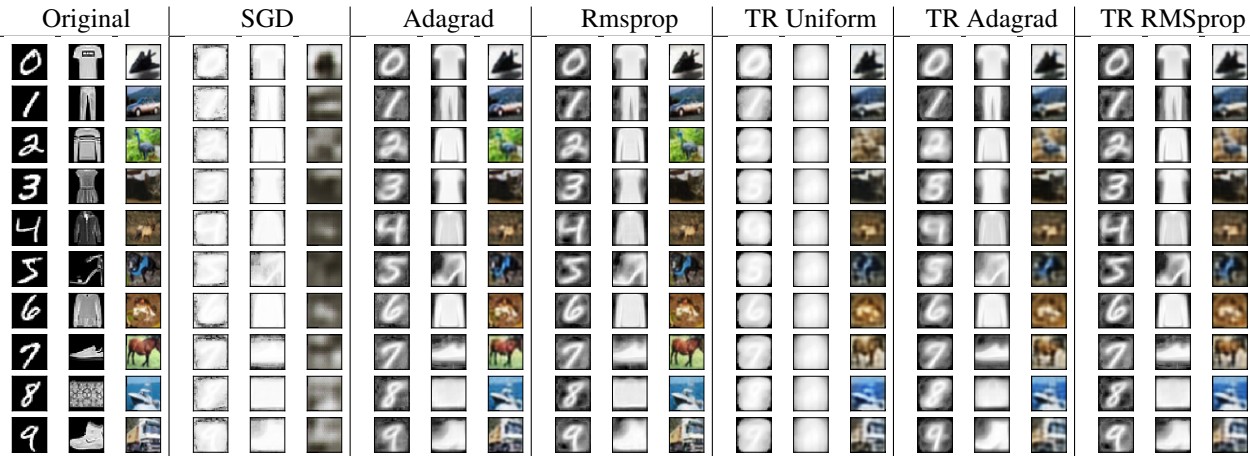

Figure 12: Original and reconstructed MNIST digits (left), Fashion-MNIST items (middle), and CIFAR-10 classes (right) for different optimization methods after convergence.

---

[12]Both datasets were accessed from https://www.tensorflow.org/api_docs/python/tf/keras/datasets

