# OpenReview forum: "Ellipsoidal Trust Region Methods for Neural Network Training"
_ICLR.cc/2020/Conference — Reject_

### Official Review · AnonReviewer3 · 2019-10-22
**Official Blind Review #3**

**Rating:** 3

**Review:**

This paper proposes ellipsoidal trust region methods for optimization on neural networks. This approach is motivated by the adaptive gradient methods and classical trust region methods. The idea of the design is reasonable, but the theoretical and empirical results are not strong. I can not support acceptance for current version.

Major comments:

1. Section 4.2 says “Algorithm 1 with A_rms ellipsoids converges with the classical rate O(\epsilon^{-2}, \epsilon^{-3}) thanks to Proposition 2 above and Theorem 6.6.8 in Conn et al.  (2000).” I have two questions about this statement.

a) What is (\epsilon^{-2}, \epsilon^{-3})? Some subscript or min/max notation looks missing.

b) I have checked Theorem 6.6.8 in Conn et al. (2000). It only shows the limit point of the sequence of iterates is a second order critical point. How to obtain the convergence rate of Algorithm 1 seems not directly. I hope the authors provide detailed derivation and present Theorem 6.6.8 here (Maybe the version of the book I have seen is not identical to yours).

2. Is there any stop criterion of the sub-problem solver? How the precision of the sub-problem affects the global convergence rate?

3. The experimental section only reports “log-loss”, which is not enough to deep learning applications. It would be interesting to validate weather the proposed method achieves lower test error than baselines.


Minor comments:

1. The values of kappa in the second and the third sub-figures of Figure 1 should be different.

2. The box of Proposition 3 is spited into page 14 and 15.

3. Equation (37) is too long and does not fit within the margins.

===================================

The authors have provided detailed derivation of the key theorem. I decide to raise my rating to 3.


**Experience Assessment:**

I have read many papers in this area.

**Review Assessment: Checking Correctness Of Derivations And Theory:**

I assessed the sensibility of the derivations and theory.

**Review Assessment: Checking Correctness Of Experiments:**

I assessed the sensibility of the experiments.

**Review Assessment: Thoroughness In Paper Reading:**

I read the paper at least twice and used my best judgement in assessing the paper.

---

> ### Author Response · Authors · 2019-11-11
> **Thank you for your review, please consider our clarifications.**
>
>
> We thank the reviewer. It seems that your more important comments are technical in nature and we have made all the changes required to address them. We provide detailed answers below and would be grateful if you could reconsider your evaluation given the clarifications we provide:
>
> 1. a) As in Section 2 - this should be $\epsilon_g$ and $\epsilon_H$. The notation is absolute standard but we have properly defined it in the revised version.
> 1. b) You are right, the paragraph was stated in a confusing manner. Thanks for pointing this out!
>
> Here is how the argument goes: (i) The Adam ellipsoids satisfy uniform equivalence [Prop. 2]. (ii) Uniform equivalence is sufficient for convergence [Conn, Theorem 6.6.8] (iii) Given convergence, a $O(\epsilon_g^{-2})$ rate can be proven for first-order- and a $O(\epsilon_H^{-3})$ rate for second-order stationarity, analogously to  [Blanchet et al. 2016] & [Gratton et al. 2017]. (As a matter of fact, the latter reference was missing in this paragraph). --> Adapting the proof for ellipsoids is straightforward, so we decided not to put it for the sake of brevity but of course we are willing to tex it granted this would influence your score positively? <--
>
> Here is a sketch: In the proof of e.g. [Gratton et al. 2017] a constant norm $\|\cdot\|$ is assumed, while we have iteration-dependent norms $\|\cdot\|_k$. Yet, simply replacing $\|\cdot\|$ with $\|\cdot\|_k$ in all of the lemmata and assuming $\|\nabla f \| > \epsilon_g^{’}$ for all k one recovers the original rate via the uniform equivalence (Prop. 2 and Def. 1) by setting $\epsilon_g=\epsilon_g’/\mu$.
>
> In summary, our approach does have guaranteed convergence and we revised the paper as stated above to make sure there is no possible confusion.
>
> 2. Yes, we use the stopping criterion defined in Eq. (36) (see l.5 of Alg.1), which ensures that each step yields (at least) a certain fraction of the model decrease of both the Cauchy- and Eigenpoint, which is sufficient to prove the convergence rates. See e.g. A3.3 in [Gratton et al. 2017] .
>
> 3. Test loss: We report training loss and the point where the methods reach 99% training accuracy on the full training set (horizontal bar) since the methods are tasked primarily to minimize the empirical risk (Eq. (1)). This is in line with many influential optimization papers published in the ML community (e.g. Kingma & Ba 2015: ADAM, Allen-Zhu 2017: Katyusha, Dauphin et al. 2014: Saddle-free Newton, Defazio et al. 2014: SAGA, Martens & Grosse 2015: K-FAC, etc.) who do also not report any results on the test set.
>
>
> Minor comments:
>
> 1. No, the two problems are specifically designed to have the same condition number in order to compare the effect of axis-alignment vs. non-axis-alignment on a fair basis.
>
> 2.&3. Thank you, we will correct these issues.

---

### Official Review · AnonReviewer1 · 2019-10-24
**Official Blind Review #1**

**Rating:** 6

**Review:**

My understanding is that the paper does not claim to deliver some great results here and now but instead suggest a promising direction ("that ellipsoidal constraints prove to be a very effective
modification of the trust region method in the sense that they constantly outperform the spherical TR
method, both in terms of number of backpropagations and asymptotic loss value on a variety of tasks").

I should say that I am not confident about my review. I believe that the authors were careful in phrasing their contribution and provided some supporting material but the wall-clock time results on resnets are too bad to envision any practical application ("advances in hardware will be needed"). It would not be a problem if the paper were the first to deal with second-order methods. However, the niche of second-order methods attempting to beat first order methods is not empty and one of the main exit plans consists in showing better wall-clock time convergence, something which is  obviously very hard.

Minor notes:
backprogations -> backpropagations

---
Update: I didn't change my score.

**Experience Assessment:**

I have read many papers in this area.

**Review Assessment: Checking Correctness Of Derivations And Theory:**

I assessed the sensibility of the derivations and theory.

**Review Assessment: Checking Correctness Of Experiments:**

I assessed the sensibility of the experiments.

**Review Assessment: Thoroughness In Paper Reading:**

I read the paper at least twice and used my best judgement in assessing the paper.

---

> ### Author Response · Authors · 2019-11-13
> **Thank you. Indeed, our main goal lies in improving and adapting Trust Region methods towards DL.**
>
> Thank you for your review. We think it’s important to stress that the purpose of our work is not to show that 2nd-order algorithms are always superior to state-of-the-art gradient methods but merely to improve and adapt existing 2nd-order algorithms to case of neural network training. On the way, we also give a geometrical interpretation of adaptive gradient methods which hopefully contributes to a better understanding of their success in deep learning.
>
> However, we think our contribution is an important step towards making 2nd-order methods practical for deep learning. In fact, we do envision that second-order methods will overtake the use of gradient methods at some point in the future (as it has been the case in many order realms of optimization such as PDE inverse problems) and our results in terms of backpropagations (Fig. 4) suggest that taking curvature into account indeed yields better update steps in modern neural net architectures such as Resnets.

---

### Official Review · AnonReviewer2 · 2019-10-26
**Official Blind Review #2**

**Rating:** 3

**Review:**

In this paper, the authors investigate the use of ellipsoidal trust region constraints for second order optimization. The authors first show that adaptive gradient methods can be viewed as first-order trust region methods with ellipsoid constraints. The authors then show that the preconditioning matrix of RMSProp and Adam can be used as norm inducing matrices for second order trust region methods. This ellipsoidal trust region method is empirically compared with first order gradient methods and spherical second order trust region methods.

Overall the paper is nicely written and very easy to read. The ideas are interesting. However, I have a number of concerns/questions about the work, that I list below.

1. Why is the preconditioning matrix of RMSProp/Adam a reasonable norm inducing matrix? One can show that the empirical Fisher is not an accurate curvature matrix in general, and so there is no reason to believe this would in fact enforce the proper ellipsoidal trust region for the method? See for example: https://arxiv.org/pdf/1905.12558.pdf.

2. I am also not convinced that Figure 2 actually shows that the curvature matrix is diagonally dominant. How do I interpret a value of 40 or 50 for this metric, and why does it imply that it is diagonally dominant?

3. The experiments also do not look very convincing to me. How sensitive is the algorithm to the hyperparameters like lambda1 and lambda2? I am also a bit confused about why different batch sizes were used for the first order gradient methods and the second order TR methods? The method overall doesn't seem to be able to match first order gradient methods, and it is not clear whether this is because of using the RMSProp/Adam preconditioner as a curvature matrix.

Given these concerns, I consider this paper to be borderline. I am happy to have a discussion with the authors and the other reviewers and change my score however.

=========================================

Edit after rebuttal:
I thank the reviewers for their response. While the updated paper has certainly improved, I think the paper still requires a much more thorough experimental evaluation. I am sticking to my score.

**Experience Assessment:**

I have published one or two papers in this area.

**Review Assessment: Checking Correctness Of Derivations And Theory:**

I assessed the sensibility of the derivations and theory.

**Review Assessment: Checking Correctness Of Experiments:**

I carefully checked the experiments.

**Review Assessment: Thoroughness In Paper Reading:**

I read the paper at least twice and used my best judgement in assessing the paper.

---

> ### Author Response · Authors · 2019-11-13
> **Thank you for your review. Please find comments below.**
>
> We are thankful for your review and the detailed comments, which we shall address in order:
>
> 1. You are right, and we are aware of this reference but answering this question comprehensively is beyond the purpose of our paper. Adaptive gradient methods have been shown empirically to be very effective at training deep neural networks and we therefore build our work on this observation instead of trying to understand the reasons behind their performance. We however agree this is an important question and we do partially address it  by giving an alternative (geometrical) interpretation (ellipsoidal constraint) but how to chose this constraint optimally is of course an open question. One answer would be curvature information but it is indeed unclear how well the Adam preconditioner approximates the PSD part of the Gauss-Newton-decomposition of the Hessian. We do believe that it contains approximate curvature information but a necessary condition for this is obviously some level of diagonal dominance in the Hessian, which we will address in 2. Another interesting explanation would be a variance reducing effect, in the sense that direction with large gradient fluctuations are “trusted less”, see https://arxiv.org/pdf/1705.07774 There, a pre-conditioner termed SVAG is proposed, which we had tried before only to find comparable results to our preconditioners.
>
> To sum up: whatever the best preconditioner may be, the main purpose of our work is to show that this “trick” can be incorporated successfully into 2nd-order algorithms.
>
> 2. Again, you are right in the sense that the absolute number is not informative. One has to take the baseline into account, which in our case is a random Wigner matrix. For these matrices the diagonal share falls as O(1/d) (Proposition 1). Of course, this baseline is somewhat arbitrary but we want to stress the fact that it is symmetric. To put this result into perspective, we added a comparison with OLS problems. As a matter of fact, one can show that the OLS Hessian has a diagonal share of sqrt(n)/d given random data. For practical batch sizes of say 32, we find that our NNs are still considerably more axis-aligned than an equally dimensioned random OLS (please see revised version of Fig. 2) for both random inputs and when training on CIFAR-10. Please let us know if this clarifies your concerns.
>
> 3. Different batch-sizes are needed because 2nd-order methods extract more information (curvature) out of the mini-batch in every step than gradient methods (this was stated in Footnote 7). Thus, they naturally need more samples in order to prevent “overfitting” a mini-batch in any given iteration. The algorithm is not very sensitive to each of the other hyperparameters individually but since there are many more than for first-order methods, we agree that more hyper-parameter tuning is needed for TR methods.
>
> Regarding the empirical result, please make sure to also take a look at Figure 3, where the ellipsoidal TR methods clearly outperform the spherical TR algorithm (purple). This is the main contribution of our paper. The SGD benchmark in Figure 4 is given in order to put these results into a realistic perspective (given current hardware etc.) and because a comprehensive benchmark of second-vs-first-order optimization has been missing in the literature. As stated in the main text, the use of ellipsoid can at least make 2nd order methods on par with 1st order methods. We believe that this is a first step towards making these methods viable for deep learning
>
>
> We hope that our comments can relieve your concerns and are looking forward to hearing your thoughts.

---

### Decision · Program_Chairs · 2019-12-19

**Decision:**

Reject

**Comment:**

This paper interprets adaptive gradient methods as trust region methods, and then extends the trust regions to axis-aligned ellipsoids determined by the approximate curvature. It's fairly natural to try to extend the algorithms in this way, but the paper doesn't show much evidence that this is actually effective. (The experiments show an improvement only in terms of iterations, which doesn't account for the computational cost or the increased batch size; there doesn't seem to be an improvement in terms of epochs.) I suspect the second-order version might also lose some of the online convex optimization guarantees of the original methods, raising the question of whether the trust-region interpretation really captures the benefits of the original methods. The reviewers recommend rejection (even after discussion) because they are unsatisfied with the experiments; I agree with their assessment.